# From Food Supplements to Functional Foods: Emerging Perspectives on Post-Exercise Recovery Nutrition

**DOI:** 10.3390/nu16234081

**Published:** 2024-11-27

**Authors:** Lifeng Wang, Qing Meng, Chun-Hsien Su

**Affiliations:** 1Public Sports Department, Xuhai College, China University of Mining and Technology, Xuzhou 221008, China; 500093.xh@cumt.edu.cn; 2School of Physical Education, Huaqiao University, Xiamen 361021, China; mq@hqu.edu.cn; 3Sport and Health Research Center, Huaqiao University, Xiamen 361021, China; 4Department of Exercise and Health Promotion, College of Kinesiology and Health, Chinese Culture University, Taipei 111396, Taiwan

**Keywords:** muscle protein synthesis, nutrient bioavailability, exercise-induced inflammation, metabolic profiling, athletic performance recovery

## Abstract

Effective post-exercise recovery is vital for optimizing athletic performance, focusing on muscle repair, glycogen replenishment, rehydration, and inflammation management. This review explores the evolving trend from traditional supplements, such as protein, carbohydrates, creatine, and branched-chain amino acids (BCAAs), toward functional foods rich in bioactive compounds. Evidence highlights the benefits of functional foods like tart cherry juice (anthocyanins), turmeric-seasoned foods, and sources of omega-3 fatty acids, including fish, flaxseeds, chia seeds, and walnuts, for mitigating oxidative stress and inflammation. Additionally, probiotics and prebiotics support gut health and immune function, which are integral to effective recovery. Personalized nutrition, informed by genetic and metabolic profiling, is examined as a promising approach to tailor recovery strategies. A systematic search across PubMed, Web of Science, and Google Scholar (2000–2024) identified studies with high empirical rigor and relevance to recovery outcomes. Findings underscore the need for further research into nutrient interactions, dosage optimization, and long-term effects on athletic performance. Integrating functional foods with personalized nutrition presents a comprehensive framework for enhanced recovery, greater resilience to physical stress, and sustained performance in athletes.

## 1. Introduction

### 1.1. Background and Importance of Post-Exercise Recovery

Post-exercise recovery is essential to athletic performance, providing a foundation for optimizing outcomes, preventing injuries, and supporting long-term health [1]. During intense physical exertion, the body experiences muscle damage, energy depletion, and fluid loss, which must be addressed in the recovery phase to restore physiological balance and prepare for future activity [2]. Recovery processes include muscle repair, glycogen replenishment, rehydration, and the reduction of inflammation, each of which contributes to physical readiness and mental resilience [3,4]. Effective recovery management promotes physical performance and mental well-being by reducing stress and decreasing the risk of overtraining, enabling individuals to maximize training benefits and sustain overall health [5,6].

A strategic recovery approach supports immediate and long-term physical readiness, reducing the likelihood of injury, enhancing athletic endurance, and enabling sustained progress in training and performance. This overview lays the groundwork for a more focused examination of nutrition’s essential role in recovery in the following section, highlighting how specific nutrients contribute to each aspect of the recovery process.

#### 1.1.1. The Role of Nutrition in Post-Exercise Recovery

Nutrition is pivotal in post-exercise recovery, offering essential elements for tissue repair, energy replenishment, and fluid balance. Protein intake is particularly crucial, as it supplies amino acids needed to repair and regenerate muscle fibers that experience microtears during exercise—a process essential for muscle adaptation and growth [1,3,7]. Sufficient protein intake supports the restoration of muscle fibers and overall muscle stability, minimizing soreness and reducing the risk of injuries, such as strains and sprains, particularly in high-impact sports [8,9,10,11]. Additionally, glycogen, depleted during strenuous activity, requires replenishment through carbohydrate intake to prepare muscles for future performance demands [12].

Antioxidants, like vitamins C and E, are integral in managing exercise-induced oxidative stress. At the same time, calcium, magnesium, and potassium are essential for maintaining muscle function and electrolyte balance, helping sustain optimal muscle performance [13,14]. Proper hydration and electrolyte replenishment are crucial in preventing dehydration and supporting sustained performance, particularly when consumed promptly. Consuming proteins and carbohydrates within the anabolic window—30 min to two hours post-exercise—can significantly enhance muscle protein synthesis and glycogen restoration, facilitating faster and more effective recovery [15,16].

Targeted nutrition protocols may further optimize recovery, with studies suggesting that branched-chain amino acids (BCAAs) can help reduce muscle soreness and fatigue, lowering the risk of injury [11,17]. Similarly, omega-3 fatty acids have been shown to promote joint health and decrease inflammation, which may help reduce overuse injuries, particularly for endurance athletes [18]. These findings emphasize that effective post-exercise nutrition requires careful attention to nutrient timing and composition to maximize recovery and minimize injury risks.

The role and necessity of dietary supplements in recovery may vary with athletic levels. Supplementation is often essential for elite athletes with rigorous training regimens to meet their elevated nutritional needs and support quick recovery and optimal performance. Their demanding schedules may preclude meeting these needs solely through diet, making supplementation valuable in maintaining a competitive edge. Conversely, amateur athletes with less intense training loads may often meet their nutritional needs through a balanced diet without relying on supplements. Recognizing these differences helps tailor nutritional recommendations that align with the specific demands of athletes across varying levels.

#### 1.1.2. Overview of Current Trends and Practices in Recovery Nutrition

Recent advancements in recovery nutrition have introduced a range of strategies to support athletes’ post-exercise recovery. This review highlights current trends and best practices widely recognized in the field. One key component is nutrient timing, specifically around the intake of protein and carbohydrates. Traditionally, consuming these nutrients within the “anabolic window”—approximately 30 min to two hours post-exercise—has been recommended to maximize muscle protein synthesis and glycogen replenishment [15,16]. This practice is particularly beneficial for elite athletes who may train multiple times a day, as immediate nutrient intake aids rapid recovery. However, recent research suggests that strict adherence to this specific timeframe may not be necessary if daily protein and carbohydrate needs are adequately met, indicating that a balanced daily intake may be more crucial for muscle repair and adaptation [19,20].

In addition to timing, the type and quality of protein consumed are vital in effective recovery. Whey protein, known for its high biological value and rapid absorption rate, is frequently recommended. However, there is growing interest in plant-based proteins for their sustainability and associated health benefits [21]. Branched-chain amino Acids (BCAAs) and essential amino acids are also commonly utilized for their benefits in muscle repair, reducing muscle soreness, and enhancing overall recovery outcomes [17]. These findings collectively support a flexible, balanced approach to post-exercise nutrition, encouraging athletes to tailor nutrient intake to their individual recovery needs and goals.

Current practices in recovery nutrition integrate nutrient timing and targeted macronutrient strategies, both of which are shown to improve recovery. By focusing on nutrient timing—particularly protein and carbohydrate intake in the immediate post-exercise period—athletes can optimize muscle protein synthesis and glycogen restoration while adapting these strategies to meet their unique training and recovery demands.

Post-exercise carbohydrate intake is essential for replenishing depleted muscle glycogen stores, with high-glycemic-index carbohydrates recommended immediately after exercise to expedite glycogen restoration [22]. Combining carbohydrates with proteins can further augment glycogen storage and muscle repair, improving recovery outcomes [23]. Adequate rehydration is also critical to recovery, as it replaces fluids lost through sweat. Sports drinks containing electrolytes such as sodium, potassium, and magnesium help maintain fluid balance and prevent dehydration, which is necessary for proper muscle function and cramp prevention [14].

Moreover, functional foods with anti-inflammatory properties, including tart cherry juice, turmeric, and omega-3 fatty acids, have been shown to reduce exercise-induced inflammation and alleviate muscle soreness [24,25]. Gut health has also emerged as a crucial aspect of recovery, with probiotics and prebiotic-rich foods incorporated into recovery diets to support gut microbiota and enhance nutrient absorption [26]. Additionally, supplements such as creatine, beta-alanine, and caffeine are frequently utilized for their performance-enhancing and recovery-boosting effects [27]. Ensuring an adequate intake of vitamins and minerals, especially those critical for energy production and muscle function, is paramount to optimizing recovery.

Technological advancements, such as mobile applications, now enable personalized nutrition plans, the tracking of dietary intake, and the provision of reminders for nutrient timing, thereby improving adherence to recovery protocols [28]. Furthermore, genetic testing and metabolic profiling innovations facilitate highly individualized nutrition strategies, allowing recovery interventions to be precisely tailored to an athlete’s unique physiological profile, maximizing the efficacy of these interventions [29].

### 1.2. Objective of the Review

This review aims to provide a comprehensive synthesis of recent advancements in post-exercise recovery nutrition, focusing on the shift from traditional dietary supplements to functional foods enriched with bioactive compounds. By examining the efficacy of various nutrients—such as protein, carbohydrates, antioxidants, and anti-inflammatory foods—in supporting muscle repair, glycogen replenishment, and inflammation reduction, we seek to clarify optimal nutritional strategies for enhancing athletic recovery and performance. For a detailed overview of the literature search and selection criteria, please refer to the Methodology section, which outlines the databases, search terms, and inclusion criteria used to capture studies published from 2000 to 2024.

## 2. Methodology

This review follows a narrative approach to synthesize current knowledge on dietary supplements and functional foods in post-exercise recovery. We utilized the following criteria for selecting sources:

### 2.1. Literature Search

A comprehensive literature search was conducted using popular academic databases, including PubMed, Web of Science, and Google Scholar. The search covered studies published in English from 2000 to 2024 to ensure that recent findings and perspectives were included. Keywords used in the search included “post-exercise recovery”, “muscle repair”, “nutrient timing”, “protein supplements”, “anti-inflammatory foods”, “electrolytes”, and “functional foods”. Boolean operators were used to combine terms effectively and refine search results.

### 2.2. Study Selection

Articles were selected based on their relevance to post-exercise nutrition and recovery outcomes to keep the review focused and relevant. Priority was given to empirical studies, systematic reviews, and meta-analyses. Articles that discussed the recovery benefits of specific nutrients, supplements, and functional foods were included. In contrast, non-peer-reviewed materials, theoretical articles, and studies without clear findings on recovery were excluded.

### 2.3. Data Extraction and Organization

Data from the selected articles were carefully extracted, with key findings categorized into themes such as protein supplementation, carbohydrate and electrolyte replenishment, anti-inflammatory effects, and emerging personalized nutrition approaches. This thematic organization allowed for an accessible summary of the leading nutritional strategies and their reported effects on recovery.

### 2.4. Quality Review

To strengthen the reliability of this review, studies were evaluated for clarity in their design and methodology, as well as relevance to post-exercise recovery. This informal quality review focused on studies that provided detailed and well-supported findings, while studies with significant methodological limitations were noted as supplementary references.

### 2.5. Limitations

The variability in study designs and populations across the included research limits this review. Additionally, while the review covers a wide range of nutritional strategies, it may not encompass all emerging trends in recovery nutrition.

By detailing the methodological approach and selection criteria, this review maintains objectivity and scientific rigor, positioning it as an evidence-based resource rather than an opinion piece. The structure reflects a critical analysis of current research, with references chosen to represent a balanced, accurate, and comprehensive view of recovery nutrition for both academic and practical applications. The main methodological limitations across cited studies are as follows: Please see Appendix A.

## 3. Overview of Post-Exercise Recovery

### 3.1. Physiological Aspects of Recovery

Muscle repair and hypertrophy are fundamental components of post-exercise recovery, especially for athletes seeking to enhance performance and reduce injury risk. Exercise, particularly resistance training or high-intensity activities, induces microtears in muscle fibers, which might initially seem detrimental but is essential for muscle adaptation and strengthening [3]. The muscle repair process involves a series of complex physiological mechanisms to reconstruct damaged fibers, thereby enhancing their resilience to future stressors.

Muscle regeneration occurs in two stages: the degeneration of the damaged fibers and a subsequent regenerative phase. Degeneration begins with the necrosis of damaged fibers, often resulting from disruptions to the sarcolemma and increased permeability of the myofibers. This stage activates inflammatory and myogenic cells, facilitating muscle repair [30]. Myogenic cells proliferate, differentiate, and fuse to repair or form new muscle fibers in the regenerative phase. This repair process is central to muscle protein synthesis (MPS), a biological mechanism through which cells produce new proteins to replace those damaged during exercise [31]. Adequate post-exercise protein intake provides the essential amino acids required for this process, with essential amino acids (EAAs) such as branched-chain amino acids (BCAAs) like leucine, isoleucine, and valine playing a significant role in stimulating MPS. Leucine activates the mTOR pathway, a major regulator of muscle protein synthesis and cell growth. Whey protein, known for its high biological value and rapid absorption, is widely regarded as an effective supplement for post-exercise recovery, supplying a rich source of EAAs and BCAAs that expedite muscle repair [32,33].

Carbohydrates are also essential for replenishing glycogen stores depleted during exercise, restoring energy levels, and supporting muscle recovery. Glycogen, stored in skeletal muscle and the liver, is the primary energy substrate during high-intensity exercise, making its restoration critical for adequate recovery and optimal subsequent performance [3]. The timely consumption of high-glycemic carbohydrates within the “anabolic window” (30 min to two hours post-exercise) is considered effective for rapid glycogen replenishment [16]. The co-ingestion of carbohydrates and protein post-exercise enhances muscle repair by promoting insulin release, facilitating amino acid uptake into muscle cells and promoting MPS. Studies suggest that ingesting a small amount of protein (0.2–0.4 g/kg/h) alongside carbohydrates (0.8 g/kg/h) can stimulate insulin release and yield glycogen repletion rates comparable to those achieved by consuming 1.2 g/kg/h of carbohydrates alone. This approach has been shown to improve subsequent exercise performance, especially for athletes engaged in multiple training or competition sessions within a short time frame [34]. Individualized glycogen replenishment strategies, tailored to exercise type and intensity, support sustained energy availability and effective recovery [12,34].

Maintaining fluid and electrolyte balance is another critical component of post-exercise recovery, necessary for optimal physiological function and preventing performance declines associated with dehydration [35,36]. Exercise results in significant losses of fluids and electrolytes, including sodium, potassium, magnesium, and calcium, through sweat, and these must be adequately replaced to prevent impairment of muscle function [37]. Hydration is key to maintaining blood volume, regulating body temperature, and ensuring proper muscle function [38]. Sports drinks containing electrolytes are recommended for prolonged or high-intensity exercise, while electrolyte-rich foods can help replenish minerals lost during exercise [39].

Following intense exercise, inflammation is triggered as a natural response to muscle fiber damage, aimed at initiating tissue repair and adaptation. This inflammatory response involves complex molecular mechanisms, including activating pro-inflammatory cytokines, oxidative stress markers, and immune responses. Key players in post-exercise inflammation include cytokines such as interleukin-6 (IL-6), tumor necrosis factor-alpha (TNF-α), and interleukin-1 beta (IL-1β), which are released by damaged muscle tissue and immune cells. This cytokine releases recruits immune cells, such as macrophages, to the injury site. Macrophages perform a dual role, initially promoting inflammation (M1 phenotype) to clear damaged cells, followed by an anti-inflammatory phase (M2 phenotype) that supports tissue repair and regeneration. This cytokine-mediated response, supported by macrophage polarization, establishes a consistent and causal mechanism that facilitates muscle repair and adaptation to repeated physical stress [2,40].

Exercise-induced oxidative stress further contributes to post-exercise inflammation. Reactive oxygen species (ROS), generated during intense physical activity, drive the inflammatory process by activating the nuclear factor kappa-light-chain-enhancer of activated B cells (NF-κB), a transcription factor regulating the expression of various inflammatory cytokines. Persistent activation of NF-κB signaling initiates a well-defined inflammatory cascade, underscoring the role of oxidative stress in the causal relationship between exercise and inflammation [41].

Evidence supporting a causal relationship, rather than a casual association, comes from studies showing that interventions targeting inflammatory and oxidative pathways can significantly modulate recovery outcomes. For instance, curcumin and omega-3 fatty acids have been shown to reduce pro-inflammatory cytokines and oxidative markers, directly impacting inflammation and aiding in muscle recovery [42,43]. This body of evidence reinforces the view that inflammation is not merely a coincidental response to exercise but a fundamental process integral to recovery and adaptation.

### 3.2. Nutritional Needs for Effective Recovery

Macronutrients—proteins, carbohydrates, and fats—are fundamental elements of an athlete’s diet, playing critical roles in energy production, muscle repair, and overall health. Understanding the specific functions of each macronutrient allows athletes to optimize their nutrition for enhanced performance and recovery.

Proteins are essential for muscle repair and growth, supplying amino acids necessary for synthesizing new muscle tissue, particularly following intense physical activity, which often induces muscle damage. Resistance exercise stimulates muscle protein synthesis (MPS) and hypertrophy, while endurance exercise triggers adaptations in mitochondrial content and oxidative capacity. Exercise increases mRNA expression, promoting protein synthesis and adaptive changes in muscle protein content [44]. Proteins are also vital for immune function, helping the body recover from injury and resist illness. Damage-associated molecular patterns (DAMPs), such as high-mobility group box 1 protein (HMGB1), are released by necrotic cells, activating immune responses that promote inflammation and tissue repair [45]. Additionally, proteins form the basis of many enzymes and hormones essential for metabolic processes and physiological function [46]. Animal proteins supply all essential amino acids necessary for muscle repair and growth, making them highly effective as a primary protein source for athletes. However, incorporating plant-based proteins can enhance dietary quality by providing additional fiber, antioxidants, and beneficial fats, which support overall wellness and may help reduce inflammation. Thus, a balanced intake of animal and plant proteins can offer a broader range of nutrients beneficial for long-term health and recovery.

Carbohydrates are the body’s primary energy source, especially during high-intensity physical activities. Stored as glycogen in muscles and the liver, carbohydrates are quickly mobilized to meet energy demands during exercise. Studies have shown that varying muscle glycogen levels through carbohydrate manipulation significantly impacts exercise performance, with higher glycogen concentrations improving performance and reducing perceived exertion during intense activities [47]. Post-exercise carbohydrate intake is essential for replenishing glycogen stores, restoring energy, and preparing muscles for subsequent training. Research indicates that carbohydrate consumption is a primary factor influencing muscle glycogen resynthesis and repeated exercise capacity, with protein supplementation further enhancing these effects when carbohydrate intake alone is insufficient [12]. Carbohydrates, including both dietary sources and stored glycogen, are the primary glucose sources utilized by the body during physical activity. Glycogen, stored in skeletal muscles and the liver, plays a crucial role in sustaining energy during high-intensity exercise, while dietary carbohydrates are essential for replenishing glycogen stores post-exercise and supporting ongoing energy needs. This dual role highlights the vital function of carbohydrates in energy metabolism and athletic performance. Under limited dietary carbohydrate intake conditions, the brain can adapt to using alternative energy sources, such as ketone bodies derived from fatty acids, to maintain its basic functions. While this adaptation allows the brain to continue functioning, it may not fully replicate the optimal cognitive performance associated with adequate glucose availability, particularly in scenarios of metabolic stress, such as in athletes with Type 1 Diabetes. In such cases, the metabolic demands of maintaining energy balance can increase significantly, potentially leading to additional physiological strain [12,47,48]. Athletes should prioritize complex carbohydrates from whole grains, fruits, vegetables, and legumes for sustained energy release and essential vitamins and minerals.

Fats play an important role in post-exercise recovery, providing a long-term energy reserve and supporting numerous physiological processes. They are crucial for cell membrane integrity and hormone production, including metabolism and reproductive health hormones. Fats also absorb fat-soluble vitamins (A, D, E, and K), which are essential for various bodily functions [36,49]. Healthy fat sources, such as avocados, nuts, seeds, olive oil, and fatty fish like salmon, supply essential fatty acids, including omega-3 and omega-6, which possess anti-inflammatory properties and support cardiovascular health [50]. Including these fats in post-exercise nutrition can reduce inflammation, support recovery, and enhance overall health. Achieving an appropriate balance of fat, protein, and carbohydrate intake based on individual needs and exercise demands is essential for optimizing recovery and long-term athletic performance [39,51].

The specific macronutrient requirements in post-exercise recovery can vary significantly based on the type of sport, as strength, power, and endurance activities each place unique demands on the body. For strength sports, such as weightlifting or bodybuilding, the primary focus is on muscle hypertrophy and repair. Protein intake is especially critical for these athletes, as it supports MPS and helps rebuild damaged muscle fibers. A daily protein intake of 1.6–2.2 g per kilogram of body weight is often recommended to meet the recovery needs of strength athletes [52].

In power sports, like sprinting or high jump, athletes rely on muscle strength and short bursts of high-intensity exertion. These athletes benefit from a balance of carbohydrates and protein to quickly replenish glycogen stores and stimulate MPS, ensuring explosive power between sessions. Carbohydrate intake within the anabolic window is essential for rapid glycogen restoration, as power sports heavily depend on muscle glycogen for peak performance. Protein also plays a critical role in sustaining muscle repair and adapting to high-force exertion, though often at slightly lower levels than required in strength sports.

Endurance sports, such as marathon running or cycling, require sustained energy output over long durations. As such, carbohydrates become the primary macronutrient focus, as high glycogen levels are necessary to maintain performance. For endurance athletes, a daily carbohydrate intake of 6–10 g per kilogram of body weight is typically recommended to maintain glycogen levels and prevent fatigue [53]. While protein is essential, it is prioritized for recovery post-exercise rather than as a primary energy source, helping to repair minor muscle damage and maintain lean body mass. A 1.2–1.6 g per kilogram daily protein intake is generally sufficient for endurance athletes. Fat intake is also relevant for endurance sports, providing a secondary energy source, especially during prolonged, moderate-intensity exercise when glycogen stores may be depleted.

These variations in macronutrient needs underscore the importance of tailored nutrition strategies for different types of athletes. Strength and power athletes emphasize protein for muscle repair and growth, while endurance athletes focus on carbohydrates for energy and recovery, with protein supporting muscle maintenance. Understanding these distinctions allows athletes in various disciplines to optimize recovery, adaptation, and overall performance.

Micronutrients, including vitamins and minerals, play critical roles in numerous physiological processes, including energy production, immune function, bone health, and muscle tissue repair [54]. Although required in smaller amounts than macronutrients, they are vital for optimizing athletic performance and overall health. Vitamins, for instance, are involved in various essential functions. Vitamin A contributes to vision, immune function, skin health, bone growth, and the maintenance of epithelial tissues [55]. B vitamins (B1, B2, B3, B6, B12, folate, biotin, and pantothenic acid) are central to energy production, as they help convert dietary energy into ATP while supporting red blood cell formation and maintaining nervous system health [56]. Vitamin C is crucial for collagen synthesis, repairing tissues such as tendons and ligaments, and functions as an antioxidant, protecting cells from free radical damage [57]. Vitamin D is essential for calcium absorption and bone health while supporting immune function and muscle strength [58]. Vitamin E is an antioxidant that protects cell membranes from oxidative stress and contributes to immune function [59]. Vitamin K is critical for blood clotting and bone metabolism [60].

Micronutrients also enhance post-exercise recovery by supporting the physiological processes necessary for repair and adaptation. Minerals such as calcium and magnesium are indispensable for muscle function, bone health, and electrolyte balance, all vital for recovery from physical exertion [61]. Iron in lean meats and whole grains is necessary for energy metabolism and red blood cell production, restoring energy levels [62]. Ensuring adequate intake of these micronutrients promotes efficient recovery, helping to minimize fatigue and support long-term performance improvements.

## 4. Role of Food Supplements in Post-Exercise Recovery

Food supplements are critical in enhancing post-exercise recovery by delivering key nutrients necessary for muscle repair, energy restoration, and overall physiological functioning. This section examines the various categories of food supplements frequently employed in post-exercise recovery, assessing their effectiveness, benefits, and potential limitations.

### 4.1. Types of Food Supplement

Food supplements are concentrated sources of specific nutrients, such as protein powders and electrolyte tablets, designed to address targeted nutritional needs when dietary intake alone may be insufficient for optimal recovery. These supplements provide isolated nutrients that complement a balanced diet, offering precise support to enhance recovery.

#### 4.1.1. Protein Supplements (Whey, Casein, Plant-Based)

Whey Protein: Whey protein is widely recognized for its rapid digestion and absorption, making it an optimal choice for post-exercise recovery. Its fast absorption allows for the swift delivery of amino acids to muscle tissues, which is particularly important during the anabolic window. Whey protein is rich in essential amino acids (EAAs), especially leucine, which is crucial in stimulating muscle protein synthesis by activating the mTOR pathway, a key regulator of cellular growth and protein synthesis essential for muscle repair and hypertrophy [63]. Furthermore, whey protein is highly versatile and can easily be incorporated into shakes, smoothies, and other recipes, providing a convenient means of boosting protein intake. It is available in various forms—concentrates, isolates, and hydrolysate—each differing in protein content and absorption rates [64].

Casein Protein: Casein, a slow-digesting protein found in dairy products, differs from whey in that it forms a gel-like substance in the stomach, resulting in a sustained release of amino acids into the bloodstream. This slow release makes casein particularly effective for prolonged muscle repair and growth, making it a suitable option for nighttime consumption to support overnight muscle recovery [65]. As the body is fasting during sleep, casein supplies a steady stream of amino acids, preventing muscle breakdown and promoting tissue repair. Casein, a slow-digesting protein, is most effective when consumed 30 to 60 min before sleep, as this timing allows for a sustained release of amino acids during the night. This prolonged release supports muscle repair and growth throughout sleep, a critical recovery period for athletes and physically active individuals [66]. Casein is also a valuable source of calcium, a mineral essential for bone health and proper muscle function [67].

Plant-based protein options, such as legumes, nuts, seeds, and whole grains, provide essential amino acids and additional nutrients like fiber, antioxidants, and healthy fats. Nuts, in particular, contribute protein and are a valuable source of plant-based nutrients that support overall health and recovery. Historically, plant-based proteins were considered inferior due to their incomplete amino acid profiles. However, advancements in food science have led to formulations that provide a complete spectrum of EAAs essential for muscle repair and overall health. Pairing different plant-based proteins can help achieve a complete amino acid profile, especially for those relying primarily on plant proteins. For example, combining legumes (such as beans or lentils) with whole grains (like rice or quinoa) creates a complementary protein source, as the amino acids missing in one food are provided by the other. Other effective pairings include hummus (chickpeas and tahini) and peanut butter on whole-grain bread [68]. Despite these improvements, plant-based proteins still face challenges in matching the efficacy of animal proteins in enhancing athletic performance. A Bayesian meta-analysis revealed that while plant-based proteins are more effective than consuming low or no protein, they are generally less effective than animal-based proteins, such as whey and casein, which provide essential amino acids that support muscle repair and recovery [69]. Although animal-based proteins like whey and casein are effective for muscle repair and recovery, they can pose allergenic risks for some individuals. Casein, particularly in its A1 form, may trigger allergies in rare cases. At the same time, lactose intolerance—due to a lack of the enzyme required to hydrolyze lactose—can also limit the use of dairy-based proteins. Plant-based proteins offer a viable alternative for these individuals, providing essential amino acids without the associated risks of dairy allergies or lactose intolerance [70,71]. These methods include chemical, biochemical, and non-thermal physical treatments designed to make plant proteins safer for consumption by sensitive populations [70]. The development of plant-based alternatives extends beyond protein supplements to include dairy analogs, such as fermented cheeses made from cashews, soy, and other plant sources, which mimic the characteristics of traditional dairy products through fermentation with specific cultures [72]. This innovation addresses dietary preferences and environmental concerns related to animal protein production, positioning plant proteins as sustainable, nutritious, and functional ingredients in modern diets [68]. While plant-based proteins have considerably improved nutritional adequacy and application, further research is necessary to overcome sensory and functional limitations and improve consumer acceptance.

In summary, protein supplements are fundamental to post-exercise recovery, with each type offering distinct advantages. Research shows that while plant-based protein supplements outperform low- or no-protein intake in enhancing athletic performance, they are generally less effective than animal-based proteins such as whey or milk in improving muscle strength and endurance [69]. However, plant proteins have been linked to a reduced incidence of metabolic syndrome, underscoring their potential health benefits beyond muscle recovery [73].

#### 4.1.2. Branched-Chain Amino Acids (BCAAs)

Branched-chain amino acids (BCAAs), comprising leucine, isoleucine, and valine, are essential amino acids that play a pivotal role in promoting muscle protein synthesis (MPS) and mitigating muscle protein breakdown (MPB), thus facilitating a net anabolic response in skeletal muscle [74]. These amino acids are essential for athletes and physically active individuals, as they enhance energy production during exercise while supporting overall muscle health. BCAAs influence the mechanistic target of rapamycin (mTOR) signaling pathway, a critical regulator of translation initiation in human muscle, which significantly impacts MPS, particularly in the post-exercise period [74]. However, despite their widespread use, the evidence regarding the efficacy of BCAA supplementation in promoting muscle hypertrophy, strength gains, and reducing post-exercise muscle soreness in resistance training remains inconclusive. Nonetheless, BCAAs may have therapeutic potential in specific medical conditions such as liver cirrhosis [75]. Additionally, research has shown that BCAAs enhance macrophage polarization, a crucial aspect of repairing exercise-induced muscle damage (EIMD), by promoting the proliferation and differentiation of muscle satellite cells through pathways such as mTORC1-HIF1α-glycolysis, underscoring their role in inflammation and muscle repair [76]. While BCAAs are integral to muscle health and energy production, their efficacy may be influenced by various factors, necessitating a personalized approach in their use for sports nutrition and therapeutic applications.

Leucine, a key component of BCAAs, is particularly effective at stimulating muscle protein synthesis by activating the mTOR pathway, which is essential for post-exercise muscle repair and growth. This corresponds with evidence suggesting that BCAA supplementation can reduce muscle soreness and fatigue, thereby lowering the risk of injuries, such as ankle sprains, in sports like basketball by promoting muscle stability and reducing soreness markers and creatine kinase levels in the blood [77]. Furthermore, BCAAs contribute to the reduction of muscle protein breakdown by decreasing the activity of proteolytic enzymes, thus preserving muscle tissue during prolonged exercise. This preservation is crucial, as BCAAs can serve as an additional energy source when glycogen stores are depleted, enhancing endurance and supporting sustained athletic performance [78]. Additionally, BCAAs have been shown to positively influence lipid and glucose metabolism, which aids in efficient energy management during physical activity [79]. Moreover, BCAAs reduce mental and physical fatigue by competing with tryptophan, lowering serotonin levels and enabling athletes to maintain higher performance. BCAAs, comprising leucine, isoleucine, and valine, are theorized to influence physical and psychological fatigue by competing with tryptophan for transport across the blood–brain barrier, thereby potentially modulating serotonin production. However, evidence from interventional studies remains inconclusive regarding the efficacy of BCAA supplementation in significantly improving fatigue or performance outcomes in athletes. While BCAAs may play a role in reducing muscle protein breakdown and supporting recovery, their direct impact on psychological fatigue and athletic performance warrants further investigation. [80]. While elevated plasma BCAA levels have been associated with the progression of atherosclerosis, these findings are derived from observational studies that do not specifically investigate the effects of BCAA supplementation. The source of plasma BCAAs in these studies could stem from dietary intake or metabolic dysregulation, independent of supplemental BCAA use. This highlights the need for further research to clarify the relationship between BCAA supplementation, metabolism, and cardiovascular health [81]. Overall, BCAAs offer a multifaceted approach to enhancing athletic performance, from stimulating muscle synthesis and providing energy to reducing fatigue. Still, they must be consumed carefully, considering their broader metabolic effects.

#### 4.1.3. Creatine Monohydrate

Creatine monohydrate is one of the most extensively studied supplements, recognized for its significant role in enhancing post-exercise recovery and supporting athletic performance. Its primary function involves increasing phosphocreatine reserves within muscle tissue, facilitating the rapid regeneration of adenosine triphosphate (ATP), the primary energy source for high-intensity physical activities. This enhancement in ATP production is critical for recovery, as it enables quicker replenishment of energy stores following intense exercise, thereby reducing fatigue and allowing athletes to return to optimal performance more rapidly [82]. Creatine supplementation has also supported recovery by minimizing muscle damage and inflammation following exhaustive physical activity, leading to faster muscle repair. This is accomplished through creatine’s anti-inflammatory effects in skeletal muscle and the brain, which aid in reducing fatigue and maintaining higher levels of spontaneous activity post-exercise [83].

Furthermore, creatine helps delay the onset of muscle fatigue, enabling longer and more intense training sessions, indirectly contributing to improved muscle recovery [84]. By promoting increased water retention within muscle cells and enhancing muscle protein synthesis, creatine creates an optimal environment for muscle repair and growth, further aiding recovery [82]. Additionally, incorporating creatine into post-exercise recovery protocols accelerates physical repair processes and may benefit cognitive recovery. This is due to creatine’s role in supporting ATP production in the brain, which may enhance mental clarity and cognitive endurance, though further research is required to substantiate these effects [85,86]. In summary, creatine is a highly effective supplement for athletes seeking to improve physical and cognitive recovery, enabling sustained performance and long-term improvement in training outcomes.

#### 4.1.4. Electrolyte Supplements

Electrolyte supplementation is essential for effective post-exercise recovery, as it aids in maintaining fluid balance, supporting nerve function, and ensuring proper muscle contractions. During exercise, particularly in hot or humid conditions, substantial amounts of electrolytes, such as sodium, potassium, magnesium, and calcium, are lost through sweat. Consequently, replenishing these electrolytes is necessary to restore physiological balance and prevent dehydration [87,88]. Sodium and chloride play critical roles in maintaining extracellular fluid balance. At the same time, potassium and magnesium are key for regulating intracellular fluid levels and are vital for optimal muscle and nerve function [87]. Calcium and magnesium are critical in muscle contraction and relaxation and essential for maintaining proper muscle function during physical activity [88,89]. Maintaining adequate electrolyte levels post-exercise supports nerve transmission, muscle coordination, and overall recovery, thereby reducing muscle soreness and fatigue while facilitating faster recovery [90]. Proper electrolyte replenishment can also help delay the onset of fatigue, enabling athletes to sustain performance during subsequent training sessions [3]. Common methods for electrolyte replenishment include sports drinks, electrolyte tablets, and powders, all of which help prevent post-exercise dehydration and prepare the body for future exertion [87]. Athletes with high sweat rates or those training in challenging environmental conditions may benefit significantly from incorporating electrolytes into their recovery protocols, ensuring sustained performance and effective recovery [88].

In addition, natural fruit-derived antioxidants, rich in polyphenols, can complement electrolyte supplementation by protecting muscle cells from oxidative damage caused by reactive oxygen species, further enhancing recovery and athletic performance [90]. However, despite the established benefits of electrolyte replenishment, knowledge regarding electrolyte, mineral, and vitamin alterations post-exercise, particularly following traumatic brain injury, remains limited. Further research is necessary to optimize recovery strategies in these contexts [87,88,89,90,91].

### 4.2. Benefits and Potential Drawbacks of Food Supplements

Food supplements play a pivotal role in addressing the nutritional needs of athletes, who often experience increased energy and nutrient demands due to their rigorous training regimens. These supplements offer numerous advantages, such as enhancing muscle repair and recovery, by providing essential amino acids through products like protein powders and Branched-Chain Amino Acids (BCAAs) [92]. BCAAs, creatine, and anti-inflammatory nutrients like omega-3 fatty acids reduce muscle soreness and inflammation, promoting quicker recovery [93]. Electrolyte supplementation during rehydration is effective for maintaining fluid balance and helping offset dehydration during subsequent intense activities, supporting sustained performance and recovery [94]. Additionally, multivitamins and minerals often address potential nutrient deficiencies, ensuring athletes receive all the essential vitamins and minerals necessary for optimal performance and health [95]. The convenience offered by food supplements is particularly beneficial for athletes with demanding schedules, as they provide an efficient means of meeting nutritional requirements without the need for extensive meal preparation [96]. Although a “food first” approach is recommended, supplements can be a practical option when dietary intake alone is insufficient to meet the nutritional demands of an athlete’s lifestyle [95]. Moreover, the use of nutritional supplements is supported by the need to enhance athletes’ adaptive responses, leading to improved performance and recovery [94]. The development of evidence-based functional foods that target muscle recovery, endurance, and strength is advancing sports nutrition innovation, underscoring the role of supplements in contemporary athletic practices [96]. While supplements should not replace a balanced diet, they remain valuable in supporting athletes’ nutritional needs and performance objectives.

Despite the benefits of food supplements in addressing specific nutritional requirements, their use also presents potential drawbacks that warrant consideration. One primary concern is the over-reliance on supplements, which can detract from consuming a balanced diet rich in whole foods that provide a broader array of nutrients and bioactive compounds that supplements cannot fully replicate [97]. Gastrointestinal issues, such as bloating, gas, or cramps, may occur with certain supplements, particularly those containing protein powders, lactose, or artificial additives. Additionally, allergic reactions to ingredients like soy or whey protein may occur, requiring careful selection of alternative protein sources [98]. The supplement industry’s regulatory limitations can lead to products with discrepancies in active ingredient amounts, sometimes containing less or significantly more than the stated dose on the label. Such variability underscores the importance of choosing third-party tested products to ensure quality and safety [80]. Furthermore, the presence of unauthorized pharmaceuticals in some food supplements, as reported by the Rapid Alert System for Food and Feed (RASFF) database, poses significant health risks, underscoring the need for a harmonized nutrivigilance system to improve safety and quality standards [99,100]. The excessive intake of certain supplements, particularly fat-soluble vitamins such as A, D, E, and K, may lead to nutrient imbalances and potential toxicity due to their accumulation in the body, emphasizing moderation and informed use [80]. Additionally, the expanding market for dietary supplements, driven by an increased incidence of lifestyle diseases and the rise of personalized nutrition, calls for advancements in nutrient delivery systems to address challenges such as poor dispersibility and instability of bioactive compounds in food matrices [101]. While supplements can be beneficial, they should be used judiciously and with a balanced diet to mitigate potential risks.

In conclusion, while food supplements can effectively support nutritional needs, enhance muscle repair, reduce soreness, improve hydration, and facilitate overall recovery, it is essential to use them strategically. Athletes should prioritize a balanced diet as the foundation of their nutritional strategy, using supplements to complement, rather than replace, whole foods. Additionally, attention to supplements’ quality, purity, and appropriate usage can mitigate potential drawbacks, ensuring they positively contribute to an athlete’s health and performance. Table 1 outlines the comprehensive role of food supplements in enhancing post-exercise recovery, focusing on their specific contributions to recovery mechanisms and overall athletic performance.

## 5. Functional Foods and Their Impact on Recovery

Functional foods offer health benefits beyond essential nutritional value, often containing bioactive compounds or added ingredients that contribute to improved health and wellness. These foods can be crucial in enhancing recovery for athletes and physically active individuals by supporting vital physiological functions and promoting overall well-being.

### 5.1. Definition and Examples of Functional Foods

Functional foods are characterized as those that offer health benefits beyond their basic nutritional content, playing a pivotal role in enhancing various physiological functions essential for recovery and long-term well-being. These foods can either naturally contain beneficial compounds or be fortified with added ingredients to improve their health-promoting properties. For example, probiotics—live microorganisms found in foods such as yogurt, kefir, and fermented vegetables—support gut health and bolster immune function by inhibiting pathogenic bacteria and enhancing the intestinal barrier [102,103]. Similarly, prebiotics, which are present in foods like chicory root, garlic, onions, and bananas, serve as substrates for beneficial gut bacteria, promoting digestive health and enhancing the diversity of the gut microbiota, which is crucial for maintaining homeostasis and disease prevention [102,104]. Antioxidant-rich foods, including berries, dark chocolate, and green tea, reduce oxidative stress and inflammation, which are vital for recovery and overall health [105]. Anti-inflammatory compounds, such as the bioactive ingredient turmeric, have been shown to support recovery by reducing inflammation without providing significant caloric or nutrient content, as whole functional foods do. This review categorizes functional foods as whole, nutrient-dense foods with health benefits (e.g., tart cherry juice). In contrast, food supplements provide isolated nutrients (e.g., standalone proteins or amino acids), and non-nutrient ingredients like turmeric, ginger, and creatine offer targeted effects without additional calories or nutrients. Functional foods also encompass fortified products, such as calcium-fortified orange juice and vitamin D-fortified milk, which promote bone health, and omega-3-enriched eggs, which support cardiovascular health [104,106]. Incorporating these foods into the diet can significantly enhance recovery by improving gut health, reducing inflammation, offering antioxidant protection, and supporting bone and cardiovascular health. This comprehensive nutritional approach not only aids athletes in optimizing their performance but also contributes to overall health and vitality, as evidenced by extensive research on functional foods [106].

### 5.2. Functional Foods for Recovery

#### 5.2.1. Dairy Products (e.g., Milk and Yogurt)

Dairy products, particularly milk, are recognized as effective functional foods that significantly aid in post-exercise recovery for athletes, owing to their comprehensive nutritional composition. Milk contains two primary proteins, whey and casein, which play critical roles in muscle repair and growth. Whey protein is rapidly absorbed, making it ideal for immediate post-exercise recovery, while casein provides a sustained release of amino acids, supporting prolonged muscle repair. These proteins supply all essential amino acids required for muscle protein synthesis, vital for promoting recovery and enhancing athletic performance [107]. Furthermore, milk is a rich source of calcium. It is often fortified with vitamin D, essential for maintaining bone density and skeletal health—critical for athletes engaging in strenuous physical activities [108]. Additionally, the electrolytes found in milk, such as potassium and sodium, help regulate fluid balance and promote rehydration, reducing the risk of dehydration following intense exercise. This is particularly important as dehydration can impair muscle performance, endurance, and strength [109].

Research has also demonstrated that milk consumption post-exercise can modulate inflammation, a key factor in recovery. Studies indicate that milk reduces the inflammatory response, as evidenced by lower concentrations of specific cytokines post-exercise than carbohydrate-based drinks [107]. Moreover, post-exercise milk consumption has been associated with reduced energy intake at subsequent meals, which may benefit athletes in managing energy balance and recovery [108]. While research on the effects of milk on intestinal health and exercise-induced inflammation suggests that milk-based beverages do not significantly mitigate exercise-related intestinal injury, they may improve work output during prolonged activities such as cycling [110]. The combination of proteins, vitamins, minerals, and electrolytes in milk underscores its value as a component of an athlete’s post-exercise nutrition strategy.

Yogurt, particularly Greek yogurt, is another highly effective functional food for post-exercise recovery. Its high protein content supports muscle repair and recovery, making it especially suitable for athletes and physically active individuals [111]. The probiotics found in yogurt, such as Lactobacillus, contribute to gut health by balancing intestinal flora, enhancing nutrient absorption, and reducing inflammation—crucial for efficient recovery [111,112]. Additionally, yogurt is rich in calcium, magnesium, and potassium, all of which support bone health and muscle function, aiding with muscle contractions and helping to prevent cramps [112]. The consumption of yogurt has been associated with a range of health benefits, including improved immune function, reduced cholesterol levels, and alleviated symptoms of lactose intolerance [112,113]. Its anti-inflammatory properties also aid in lowering obesity-induced inflammation and improving glucose metabolism, further supporting recovery and overall health [114,115].

Incorporating dairy products such as milk and yogurt into post-exercise nutrition provides athletes with essential nutrients that promote muscle repair, bone health, and rehydration. These versatile foods can be easily incorporated into smoothies, shakes, or snacks, offering convenient, nutrient-dense options to optimize recovery strategies.

#### 5.2.2. Anti-Inflammatory Foods (e.g., Tart Cherry Juice, Turmeric, and Other Fruit Juices)

Anti-inflammatory foods support athletic recovery by reducing exercise-induced inflammation and muscle soreness, promoting overall health. Tart cherry juice and turmeric are particularly notable for their potent anti-inflammatory properties. Tart cherry juice is rich in anthocyanins, powerful antioxidants that help mitigate oxidative stress and inflammation resulting from intense physical activity. This effect is consistent with findings from other anthocyanin-rich fruits, such as blueberries, which have been shown to elevate post-exercise levels of anti-inflammatory oxylipins, aiding in inflammation resolution and muscle recovery [116]. Polyphenols in various fruits and vegetables, including those found in tart cherry juice, are recognized for their antioxidant and anti-inflammatory effects. However, their efficacy may vary depending on the specific polyphenol and its application [117]. The consumption of natural fruit-derived antioxidants, such as those found in tart cherry juice, is recommended as a safe nutritional approach to protecting muscle cells from excessive reactive oxygen species (ROS) and alleviating delayed-onset muscle soreness (DOMS) [90].

In the case of bromelain, an enzyme found in pineapple juice, findings vary similarly. Although bromelain has been promoted for its anti-inflammatory effects, research provides mixed results. For example, one study reported reduced perceived exertion but no significant effect on muscle damage markers following protease supplementation, including bromelain, during consecutive days of cycle racing. Similarly, another study found no significant difference in managing delayed onset muscle soreness with bromelain compared to ibuprofen. These studies highlight the need for further research to establish more definitive conclusions about bromelain’s efficacy in recovery [118,119,120]. Additionally, while not directly related to tart cherry juice or turmeric, marine bioactive compounds have been noted for their anti-inflammatory properties, underscoring the broader potential of natural compounds in functional foods to support recovery and overall health [5]. By incorporating anti-inflammatory foods such as tart cherry juice, turmeric, and other fruit-based juices into their diets, athletes can significantly enhance recovery, reduce inflammation, improve muscle repair, and support overall athletic performance.

#### 5.2.3. Omega-3 Fatty Acids (e.g., Fish, Flaxseeds, Chia Seeds, and Walnuts)

Omega-3 fatty acids, known for their anti-inflammatory properties, are available from various sources. While fish oil is a popular choice, other sources such as flaxseeds, chia seeds, and walnuts provide plant-based Omega-3s (specifically ALA), making it accessible to individuals on plant-based diets. Including various sources can help meet Omega-3 needs and support overall health. These fatty acids play a crucial role in modulating the inflammatory response by reducing the production of pro-inflammatory cytokines and eicosanoids, thereby minimizing muscle soreness and stiffness, and promoting faster recovery following intense physical exertion [100]. Research has demonstrated that omega-3 supplementation can enhance muscle regeneration and improve physical performance, making it particularly beneficial in managing conditions such as sarcopenia by increasing muscle strength and endurance [121]. In addition to their role in muscle recovery, omega-3 fatty acids also support cardiovascular health by lowering triglyceride levels, reducing blood pressure, improving circulation, and enhancing nutrient delivery to muscles during recovery [122]. In older adults, omega-3 supplementation, combined with resistance exercise training, has increased resting metabolic rate, boosted fatty acid oxidation, and reduced systemic inflammation, further supporting recovery and performance [123]. Moreover, omega-3 fatty acids contribute to joint health by alleviating pain and stiffness, providing additional benefits for athletes involved in high-impact activities [124].

For optimal intake, athletes can obtain omega-3s through fish oil supplements or by consuming omega-3-rich foods, such as salmon, mackerel, and sardines, while vegetarians and vegans can opt for algae oil as a plant-based source of DHA. Regular consumption of omega-3s may support faster recovery by reducing delayed-onset muscle soreness (DOMS) and markers of muscle damage. However, evidence regarding their effects on maximal muscle strength recovery and athletic performance is mixed, with some studies reporting no significant improvements in performance outcomes [121]. Table 2 highlights the comprehensive role of functional foods in supporting post-exercise recovery, emphasizing their multifaceted contributions to optimizing recovery outcomes.

## 6. Comparing Food Supplements and Functional Foods

### 6.1. Nutritional Efficacy

#### 6.1.1. Protein and Muscle Recovery

Both food supplements and functional foods are integral to post-exercise muscle recovery, primarily through their provision of protein. Protein supplements, such as whey and casein, offer a concentrated and rapidly absorbed source of essential amino acids, critical for muscle protein synthesis and repair. Whey protein, mainly, is highly effective when consumed during the “anabolic window” following exercise due to its high biological value and rapid absorption rate. Adding protein to the carbohydrate supplement can further enhance glycogen synthesis, promote protein synthesis, reduce muscle damage, and stimulate muscle tissue repair and adaptation [16,125,126,127,128].

In contrast, functional foods, such as dairy products (e.g., milk and yogurt), offer comparable benefits by providing a natural source of protein that supports muscle repair over an extended period. This is primarily due to the slower digestion rate of casein, which allows for a sustained release of amino acids, facilitating prolonged muscle recovery [50,91,92,95]. Moreover, functional foods often contain additional nutrients, such as calcium and vitamin D, essential for bone health and contribute to long-term recovery [89,93].

#### 6.1.2. Anti-Inflammatory and Oxidative Stress Management

While vitamins C and E are commonly known for mitigating oxidative stress, some research suggests that excessive antioxidant supplementation may interfere with exercise-induced adaptations by overly buffering reactive oxygen species (ROS), critical in signaling pathways for muscle adaptation [13,90,129]. These compounds play a critical role in counteracting the damage caused by free radicals, thereby facilitating more efficient recovery.

Similarly, functional foods such as tart cherry juice, turmeric, and polyphenol-rich fruits offer comparable anti-inflammatory effects, though they are typically consumed as part of a regular diet [24,25,116,117]. The key advantage of functional foods in this context is their comprehensive nutritional profile, which addresses inflammation and provides additional vitamins, minerals, and fiber, contributing to overall health and well-being beyond the scope of inflammation management. 

### 6.2. Practicality and Usage

#### 6.2.1. Convenience of Food Supplements

Food supplements are often preferred for convenience, particularly by athletes with elevated nutrient demands and demanding schedules. These supplements provide a quick and efficient means of achieving specific nutritional targets, such as adequate protein, electrolytes, or creatine intake, which may be difficult to obtain solely through whole foods [15,92]. However, there are concerns that over-reliance on supplements could lead to neglecting a balanced diet rich in whole foods, which offer a broader range of nutrients and associated health benefits [130].

#### 6.2.2. Sustainability of Functional Foods

In contrast, functional foods promote a more holistic approach to nutrition, contributing to both recovery and long-term health through consistent dietary consumption. These foods support various aspects of health, such as digestive health through probiotics and cardiovascular health via omega-3 fatty acids, while also offering anti-inflammatory benefits. This makes functional foods a sustainable and advantageous option for both athletes and the general population [37,102,104,106,121]. However, their effects may be less immediate and targeted than food supplements, specifically formulated to meet particular nutritional needs in concentrated forms.

### 6.3. Risks and Limitations

#### 6.3.1. Over-Reliance on Supplements

Although supplements offer significant benefits, excessive use can result in adverse effects, including gastrointestinal discomfort or nutrient imbalances, particularly with fat-soluble vitamins such as A, D, E, and K [98,101]. Moreover, the lack of rigorous regulatory oversight within the supplement industry raises concerns regarding product purity and quality, posing potential consumer risks [131].

#### 6.3.2. Functional Foods and Accessibility

While generally safer and less susceptible to misuse, functional foods may not always deliver the specific nutritional support required by athletes involved in high-intensity training [132,133]. Furthermore, the natural variability in nutrient composition and slower absorption rates can limit their effectiveness when immediate recovery is critical [15,134]. Table 3 comprehensively compares supplements and functional foods, emphasizing their nutritional efficacy, practicality, usage, and associated risks and limitations.

Functional foods and supplements are beneficial in supporting recovery and enhancing athletic performance, offering nutrients that aid in muscle repair, inflammation reduction, and rehydration without necessarily being required for survival or growth. While advantageous, these supplements should not displace whole foods that provide a full spectrum of essential nutrients necessary for a balanced diet. In contrast, functional foods provide a sustainable approach to recovery and long-term health through bioactive compounds like anti-inflammatory agents and probiotics. Athletes should adopt a balanced strategy for optimal performance and well-being, using supplements to meet acute demands and functional foods to support ongoing recovery and overall health. This approach ensures the body receives both immediate and long-term nutritional support.

## 7. Future Directions and Emerging Perspectives

### 7.1. Innovations in Recovery Nutrition

#### 7.1.1. Personalized Nutrition Plans Based on Genetic and Metabolic Profiling

The future of recovery nutrition is evolving towards a more personalized approach, driven by advancements in genomic and metabolic testing that enable the creation of individualized nutrition plans. Personalized nutrition considers an individual’s unique genetic and metabolic profiles, optimizing health and performance by addressing specific dietary needs and potential deficiencies that traditional assessments might overlook [135,136]. This development is part of a broader shift towards personalized medicine, emphasizing individual differences’ importance in health management [137]. By utilizing biomarkers and performance assessments, tailored health plans can be developed to help individuals effectively achieve their wellness goals [138]. Metabolomics plays a crucial role in this process by providing insights into unique metabolic profiles and facilitating the design of nutrition strategies tailored to an individual’s specific requirements [136]. As precision nutrition advances, it promises to improve recovery outcomes by ensuring dietary recommendations are scientifically grounded and customized to each individual’s needs [139]. This approach significantly advances how nutrition is applied to recovery contexts.

(1)Genetic Profiling: Genetic testing is critical in identifying individual variations in nutrient metabolism, which can significantly influence dietary recommendations for optimal health. Integrating genetics and metabolomics into personalized nutrition strategies enables the identification of single nucleotide polymorphisms (SNPs) that affect nutrient metabolism and health outcomes, providing a comprehensive approach to personalized dietary interventions [137]. For instance, genetic variants in the vitamin D receptor (VDR) gene may influence metabolic pathways, impacting conditions such as type 2 diabetes and metabolic syndrome. However, the full extent of these effects remains under investigation [140]. Moreover, genotype-based nutritional supplementation has demonstrated that genetic variations can alter the metabolism of nutrients such as vitamin D, iron, and calcium, underscoring the importance of personalized nutrition for improving nutrient absorption and utilization [141]. In nutrigenomics, vitamin D serves as a key example, with its interaction with the VDR modulating gene expression across various tissues, influencing metabolism, bone formation, and immune function [142]. Nutrigenomics further explores how genetic variations influence nutrient-mediated pathways, offering insights into optimizing health and preventing disease through tailored dietary practices [143]. Understanding genetic differences is essential for developing personalized nutrition plans that cater to individual metabolic needs, enhancing nutrient absorption and health outcomes.(2)Metabolic Profiling: Metabolic profiling involves analyzing an individual’s metabolic responses to various foods and nutrients. The concept of metabotyping, which groups individuals based on their metabolic profiles, has improved dietary quality and lipid profiles compared to generalized dietary advice [136,144]. This approach, supported by omics technologies such as metabolomics, provides detailed insights into metabolism by measuring metabolites that reflect food intake and the effects of diets on endogenous metabolism [135]. These technologies enable the identification of clinically relevant subgroups or metabotypes, allowing for personalized dietary recommendations that may improve adherence to healthier diets and health outcomes [135,144]. By understanding these metabolic responses, customized nutrition plans can be developed to optimize recovery and overall health.(3)Microbiome Analysis: The gut microbiome is vital in nutrient absorption, immune function, and inflammation regulation. Microbiome-based therapies, such as fecal microbiota transplantation and probiotics, are essential for managing lower gastrointestinal diseases by regulating gut microbiota and microbial compounds that influence disease development and immune responses [145]. These findings highlight the importance of incorporating microbiome analysis into personalized nutrition plans, which can recommend appropriate probiotics and prebiotics to support gut health and enhance recovery.

#### 7.1.2. Advances in Food Technology and Supplement Formulation

Advancements in food technology and supplement formulation are driving innovation in recovery nutrition, enabling the creation of more effective, bioavailable, and convenient products tailored to the needs of athletes and active individuals.

(1)Enhanced Bioavailability: One of the key challenges in nutrition is ensuring that the body effectively absorbs and utilizes consumed nutrients. Nanotechnology facilitates the encapsulation of bioactive agents in colloidal delivery systems designed to protect and deliver nutrients in a bioavailable form, enhancing personalized nutrition products tailored to individual needs such as genetics and lifestyle [146]. Lipid-based nanocarriers, including liposomes and niosomes, improve the intestinal absorption of nutraceuticals by solubilizing them in the intestinal environment and facilitating lymphatic transport, thereby addressing challenges related to poor bioavailability and degradation in the gastrointestinal tract [147]. Nanotechnology in nutrient delivery systems also addresses issues related to the solubility and stability of nutraceuticals, such as curcumin and vitamins, improving their oral bioavailability and reducing first-pass metabolism [148]. Furthermore, nanoparticle technology is being explored for micronutrient fortification in food crops, such as vitamin B12, to combat deficiencies and enhance nutrient delivery without genetic modification, contributing to a more sustainable food system [149]. These innovations improve nutrient delivery efficacy while reducing the risks associated with higher doses, marking a significant advancement in nutrition and health care [150].(2)Functional Ingredients: New functional ingredients are being developed and incorporated into food and supplements to enhance recovery. Plant-based functional foods, such as those containing soy protein, are gaining attention for their potential to provide complete amino acid profiles and reduce inflammation and oxidative stress, both of which are critical in preventing metabolic diseases like diabetes and obesity [151,152]. Soybean proteins, including peptides like lunasin, have demonstrated promising antioxidant and immunomodulatory properties, suggesting their role in functional foods promoting health and preventing oxidative stress-related diseases [152]. Additionally, incorporating plant-based proteins and bioactive compounds into functional foods addresses the rising need to prevent lifestyle-related diseases and promote overall well-being [153]. Novel antioxidants, such as those found in tart cherry and omega-3 fatty acids, have been identified as effective in reducing exercise-induced muscle damage (EIMD) by mitigating oxidative stress and inflammation, thereby enhancing post-exercise recovery [154]. As the market for functional foods expands, continued research and innovation are essential to validate health claims and develop consumer-focused products that meet the growing demand for health and wellness solutions [153].

#### 7.1.3. Practical Implications of Personalized Nutrition

The shift towards personalized nutrition in sports has significant implications for enhancing athletic performance, recovery, and long-term health by tailoring dietary strategies to individual needs. Personalized nutrition considers genetic, metabolic, and physiological differences, allowing for precise dietary recommendations that align with each athlete’s profile. Genetic testing, for instance, can identify predispositions to specific nutritional needs or sensitivities, such as caffeine metabolism, enabling practitioners to adjust dietary plans accordingly [155]. Metabolic profiling offers detailed information on an athlete’s macronutrient requirements, facilitating targeted protein, carbohydrate, and fat intake adjustments that align with energy expenditure and training demands [19]. For endurance athletes, higher carbohydrate intake is crucial for sustained performance, while strength athletes benefit from increased protein to support muscle growth [19]. Personalized nutrition also incorporates nutrient timing to maximize recovery, such as consuming carbohydrates post-exercise for glycogen replenishment and protein before sleep to promote muscle protein synthesis [19]. Addressing food sensitivities and digestive health, personalized nutrition can reduce gastrointestinal distress, which is particularly relevant for endurance athletes [155]. This approach supports efficient nutrient absorption, which is essential for sustained performance. Beyond physical benefits, personalized nutrition enhances adherence by respecting athletes’ cultural, ethical, and personal preferences, fostering a supportive environment that empowers athletes to maintain consistent nutrition plans [138]. Integrating genetic, metabolic, and lifestyle data enables the creation of dynamic nutrition plans that evolve with an athlete’s goals and performance needs, positioning personalized nutrition as a valuable tool for optimizing recovery and peak performance [156]. Advances in food technology, such as nanotechnology, further improve nutrient bioavailability, making recovery supplements more effective [155]. Collaboration among athletes, coaches, and nutritionists, supported by real-time data, underscores the potential of personalized nutrition in optimizing recovery and sustaining peak performance [96].

### 7.2. Potential Research Areas

#### 7.2.1. Identifying New Functional Foods with Recovery Benefits

The exploration and identification of novel functional foods with potential recovery benefits represent a promising area for future research. Scientists continue to discover and investigate foods rich in bioactive compounds that promote athletic recovery and overall health [157]. Emerging superfoods such as moringa, spirulina, and baobab may serve as new sources of essential nutrients and antioxidants, offering recovery benefits through their ability to reduce inflammation, enhance immune function, and support muscle repair [158]. Studying traditional foods from various cultures—such as medicinal plants and herbs utilized in Ayurveda, Traditional Chinese Medicine, and Indigenous healing practices—may reveal unique bioactive compounds contributing to modern nutritional strategies [159]. Focused research on specific bioactive compounds, such as polyphenols, flavonoids, and peptides, can help elucidate their roles in recovery. Understanding how these compounds interact with physiological processes at the molecular level may lead to the development of targeted functional foods.

#### 7.2.2. Long-Term Effects of Regular Use of Food Supplements and Functional Foods

Investigating the long-term effects of regular use of food supplements and functional foods is essential to understanding their sustained impact on health and performance. While numerous studies have demonstrated the short-term benefits of certain functional ingredients in foods or supplements, these findings are not universally established across all ingredients, and evidence for long-term benefits remains limited. Many studies focus on short-term effects, as these are more feasible, leaving gaps in understanding the sustained impact of functional ingredients on health and performance [160]. These studies could determine whether the regular use of specific supplements or functional foods reduces the risk of chronic conditions such as cardiovascular disease, diabetes, and osteoporosis, ultimately contributing to prolonged athletic careers and improved health. Furthermore, research into the long-term effects on nutrient absorption and utilization is necessary to identify potential issues such as nutrient imbalances or deficiencies, particularly concerning fat-soluble vitamins and minerals that may accumulate in the body over time. Understanding whether the benefits observed from short-term use are sustained with prolonged consumption, as well as examining any potential adaptations in the body’s response, is crucial. Evaluating long-term supplement use’s safety and possible side effects will help refine dosage recommendations and identify risks associated with prolonged intake.

#### 7.2.3. Interactions Between Different Nutritional Strategies

Research into the interactions between various nutritional strategies is critical for optimizing recovery and performance. This area of study focuses on how different dietary components and the timing of intake influence one another and affect the overall efficacy of nutritional interventions. Investigating the synergistic effects of combining supplements and functional foods may reveal combinations that enhance recovery. For example, combining omega-3 fatty acids with antioxidant-rich foods could amplify their anti-inflammatory benefits [37,83,91]. Research into nutrient timing—such as pre-workout, intra-workout, and post-workout consumption—can offer insights into the optimal timing for specific foods and supplements to maximize recovery [15,16,28]. Furthermore, examining how dietary patterns, such as ketogenic, vegetarian, or intermittent fasting, interact with supplements and functional foods will allow customized nutrition plans to suit individual needs and preferences better. Understanding how the presence of certain foods or nutrients affects the bioavailability and absorption of others is also critical. For instance, research into how dietary fats influence the absorption of fat-soluble vitamins could lead to more effective supplementation protocols.

## 8. Expanded Discussion on Long-Term Effects in Recovery Nutrition

Understanding the long-term effects of recovery nutrition is essential for developing sustainable and effective dietary strategies for athletes. While immediate recovery benefits, such as enhanced muscle repair and glycogen replenishment, are well documented, the prolonged use of specific supplements and functional foods raises questions about nutrient balance, safety, and the potential for adverse effects. Ensuring recovery practices contribute positively to an athlete’s career requires carefully balancing nutrient intake, monitoring for potential side effects, and ongoing research.

### 8.1. Nutrient Balance and Long-Term Health

An optimal nutrient balance is crucial to avoid potential deficiencies or excesses that could compromise health. For instance, high protein intake, often recommended for muscle repair, must be balanced with adequate carbohydrates and fats to ensure comprehensive nutritional support [19]. Overemphasis on protein without sufficient carbohydrate intake can lead to impaired glycogen recovery, ultimately impacting performance and muscle endurance in the long term. Additionally, a balanced intake of micronutrients such as calcium, magnesium, and vitamins D and C is essential to support bone density, immune function, and antioxidant defense, particularly for athletes who engage in high-impact sports [161]. Long-term reliance on specific macronutrient distributions should be carefully evaluated to prevent imbalances that may negatively affect metabolic and cardiovascular health.

### 8.2. Potential Side Effects of Prolonged Supplement Use

Prolonged use of certain supplements, while beneficial in the short term, may have adverse effects over extended periods. For example, long-term protein supplementation, especially at high doses, can place a strain on the kidneys and may increase the risk of kidney dysfunction in individuals with pre-existing renal conditions [162]. Similarly, although anti-inflammatory, high-dose omega-3 fatty acid supplementation may have blood-thinning effects that could lead to complications if not monitored [163]. Antioxidant supplements, such as high vitamin C or E doses, could potentially interfere with the body’s natural adaptive responses to exercise-induced oxidative stress, possibly diminishing training adaptations over time [164]. Athletes and practitioners should monitor and adjust supplement intake based on individual health markers and training cycles, avoiding continuous high-dose supplementation unless indicated.

### 8.3. The Need for Longitudinal Studies in Recovery Nutrition

Current research on recovery nutrition largely focuses on short-term effects, leaving a gap in understanding the long-term outcomes associated with sustained dietary and supplement use. Longitudinal studies are essential to examine how chronic use of recovery supplements affects athletes over their lifespan, including potential impacts on metabolic health, cardiovascular function, and the risk of chronic diseases. For example, studies tracking the long-term impact of high-protein diets on kidney health and bone density or the effects of long-term antioxidant supplementation on cellular adaptations could provide crucial insights [165]. Additionally, examining the effects of personalized nutrition over time, mainly through genetic and metabolic profiling, could help determine whether individualized approaches yield sustained benefits in performance and health outcomes [166].

### 8.4. Sustainability and Habit Formation

Long-term recovery strategies must also consider sustainability and habit formation, as overly restrictive or complex nutritional practices may be problematic for athletes to maintain [167]. Developing recovery protocols that are easy to adhere to, flexible, and adaptable to an athlete’s changing needs throughout their career will promote consistency. Encouraging diverse, nutrient-dense food choices and limiting reliance on supplements can help athletes develop balanced dietary habits that support long-term health. Exploring the cumulative effects of whole-food-based recovery practices compared to supplement-heavy approaches will provide insights into which methods foster lasting wellness and performance.

## 9. Conclusions

Nutrition is fundamental to post-exercise recovery, influencing muscle repair, inflammation reduction, immune function, and overall athletic performance. This review highlights the roles of food supplements and functional foods in enhancing recovery outcomes. Proper nutrient intake replenishes glycogen, repairs muscle tissue, supports the immune system, and reduces inflammation, with the timing and quality of nutrients being critical to maximizing benefits. Supplements, such as protein for muscle repair, omega-3 fatty acids for anti-inflammatory effects, and electrolytes for hydration, provide targeted support when dietary intake may be insufficient. Functional foods like tart cherry juice, turmeric, and leafy greens offer antioxidants, promote gut health, and contribute to recovery through bioactive compounds.

The field is advancing towards personalized nutrition, driven by genetic and metabolic profiling, which enables tailored nutrition plans based on individual needs and metabolism. Genetic testing can reveal unique nutrient requirements, while metabolic profiling supports precise macronutrient adjustments. Age-specific strategies also play a role, with young athletes requiring higher caloric intake and nutrient-dense foods to support growth and master athletes benefiting from protein-rich diets, antioxidants, and omega-3s to counteract age-related changes and support recovery. Proper hydration, especially electrolyte management, remains essential across all age groups.

Future trends in food technology and supplement formulation, such as nanotechnology for enhanced bioavailability, are promising for improving nutrient delivery and efficacy. Continued research into nutrient timing, long-term effects, and the synergistic benefits of combining functional foods with targeted supplementation will further refine recovery nutrition protocols. Tailoring recovery strategies to individual and age-specific needs allows athletes to optimize performance, support long-term health, and achieve peak recovery outcomes.

## Figures and Tables

**Table 1 nutrients-16-04081-t001:** Comprehensive role of food supplements in post-exercise recovery.

Supplement Name	Supplement Key Features	Benefits and Potential Drawbacks	Contraindications/Side Effects
Protein Supplements (Whey, Casein, Plant-Based)	Rapid digestion (whey), sustained release (casein), plant-based options for complete amino profiles	Enhances muscle repair and hypertrophy; convenience; potential bloating or allergies with whey [63,64,65]	Allergic reactions (soy, whey); gastrointestinal issues [98]
Branched-Chain Amino Acids (BCAAs)	Includes leucine, isoleucine, and valine; readily used as energy in muscle	Reduces muscle soreness and fatigue; supports endurance; risks with excess intake [77,78]	Potential contribution to atherosclerosis if overconsumed [81]
Creatine Monohydrate	Enhances ATP production; aids high-intensity recovery	Minimizes muscle damage and inflammation; supports muscle growth and cognitive clarity [82,83]	May cause water retention kidney stress with long-term misuse [86]
Electrolyte Supplements	Supports fluid and electrolyte balance (sodium, potassium, magnesium)	Prevents dehydration; supports nerve function and recovery in hot conditions [87,88]	Excessive sodium intake could affect blood pressure [91]

**Table 2 nutrients-16-04081-t002:** Comprehensive role of functional foods in post-exercise recovery.

Functional Food Names	Functional Food Key Features	Benefits and Potential Drawbacks	Recommended Dosage
Dairy Products (e.g., Milk, Yogurt)	Rich in proteins (whey, casein) and electrolytes, it supports muscle and bone health	Enhances muscle repair, provides sustained amino acid release, supports rehydration; may not suit lactose-intolerant individuals [107,108,111]	250–500 mL post-exercise [109]
Anti-inflammatory Foods (e.g., Tart Cherry Juice, Turmeric)	High in antioxidants and anti-inflammatory compounds	Reduces muscle soreness, mitigates oxidative stress, and may cause gastrointestinal discomfort in excess [116,117]	Tart cherry juice: 8–12 oz; Turmeric: 1–2 g daily [90,118]
Omega-3 Fatty Acid-rich Foods (e.g., Fish, Flaxseeds)	Anti-inflammatory, supports cardiovascular health	Minimizes exercise-induced inflammation, supports muscle recovery, and reduces joint pain; excessive intake may thin blood [121,122]	1–2 g daily of combined EPA/DHA for optimal effect [100]

**Table 3 nutrients-16-04081-t003:** Comparing food supplements and functional foods.

Aspects	Food Supplements	Functional Foods
Nutritional Efficacy	It provides concentrated sources of specific nutrients (e.g., protein, BCAAs) that are ideal for rapid muscle repair and recovery [63,92,93,126].	Delivers a natural, comprehensive nutrient profile supporting long-term health through whole foods [24,25,102,104,117]. Functional foods offer antioxidants for recovery, but supplements are often needed to achieve effective nutrient doses.
Practicality and Usage	Convenient and dose-controlled, making it suitable for athletes needing quick nutrient intake [15,92,130]. Food supplements are portable and shelf-stable, offering convenient nutrition for athletes, while functional foods are often perishable and require careful storage.	Integrates naturally into daily meals, offering holistic benefits but may require more preparation [37,106,121].
Risks and Limitations	Overuse risks include gastrointestinal issues, possible regulatory oversight gaps, and the risk of nutrient imbalance [80,98,100,101].	Generally safer and more accessible, but may not deliver the same immediate effects for intensive recovery [132,133,134].
Indicative Costs	Often higher due to processing and quality control [15,92].	Typically more affordable, being part of regular dietary intake [37,102,106].

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
