# Peer review of "From Food Supplements to Functional Foods: Emerging Perspectives on Post-Exercise Recovery Nutrition"

_nutrients, 2024, doi:10.3390/nu16234081_

Round 1
Reviewer 1 Report
Comments and Suggestions for Authors
This is a well written article, it is extensive. Its almost like a book chapter. I wonder if the journal will want something this long?
Specific items
L59: should have citations. I would be curious to know what studies show the ability to reduce injury risk
L65: what is meant by widely accepted methodologies. I read that and think - research methodologies. What does that phrase mean in relation to contemporary trends?
L177: wouldn't animal based provide all amino acids? Why would you need plant? Or what is meant by a balanced intake of both? Animal alone should supply what you need; any plant based is extra
L191: I think this is considered standard; however, what about those who do a low-carb, keto, or carnivore diet. Is there any role for using fatty acids and conversion to glucose in the body to help with brain function? It seems like glucose is the key vs. carbs?
L258: so when would one need to consume casein in order to utilize it during sleep?
L262: are nuts considered a plant based option?
L266: can you provide some examples? Is this supposed to relate to pairing of proteins?
L354: try to avoid 1 sentence paragraphs
L538: can you also get Omega 3 from other sources? If so, I wouldn't recommend having (e.g., fish oil). It would suggest that that is a primary or only source
Author Response
Based on Reviewer's Comments: Point-by-Point Response
Below is a point-by-point response addressing each of the reviewer's comments. All revised text is highlighted in red. You may refer to the modified version, with updated sentences marked in yellow for easy reference.
Thank you.
Comments and Suggestions for Authors
- reviewer 1
This is a well written article, it is extensive. Its almost like a book chapter. I wonder if the journal will want something this long?
Response:
We sincerely appreciate your positive feedback and thoughtful observations regarding the manuscript’s length. We intended to create a thorough review that could provide a comprehensive resource on the evolving role of dietary supplements and functional foods in post-exercise recovery, particularly in light of the shift from traditional strategies toward more personalized approaches.
Given the broad scope of this topic, we recognize that the manuscript may seem extensive. However, we have positioned it as a general literature review to cover both foundational insights and recent advancements in recovery nutrition. We hope this depth aligns with the journal’s aim to offer readers a well-rounded, evidence-based perspective. We would be grateful if the manuscript could be considered for publication in its revised form.
Specific items
L59: should have citations. I would be curious to know what studies show the ability to reduce injury risk
Response:
Thank you for your valuable feedback. We appreciate your suggestion to provide citations supporting the statement on reducing injury risk. In the revised manuscript, we have included relevant studies that discuss the role of targeted nutrients in reducing injury risk, mainly related to muscle stability, soreness reduction, and enhanced recovery protocols.
Revised Text:
Targeted nutrition protocols may further optimize recovery, with studies suggesting that branched-chain amino acids (BCAAs) can help reduce muscle soreness and fatigue, lowering the risk of injury [11,17]. Similarly, omega-3 fatty acids have been shown to promote joint health and decrease inflammation, which may help reduce overuse injuries, particularly for endurance athletes [18]. (please see page 2, lines 65-69)
L65: what is meant by widely accepted methodologies. I read that and think - research methodologies. What does that phrase mean in relation to contemporary trends?
Response:
Thank you for your observation regarding the phrase “widely accepted methodologies.” We agree that this could be more clearly stated to avoid ambiguity. In this context, "widely accepted methodologies" refer to standard practices and strategies in recovery nutrition broadly endorsed within sports science, such as nutrient timing and the prioritization of specific macronutrients for recovery. In the revised manuscript, we have clarified this term to more accurately reflect its application to contemporary nutritional strategies rather than research methodologies.
Revised Text:
Recent advancements in recovery nutrition have introduced a range of strategies to support athletes' post-exercise recovery. This review highlights current trends and best practices widely recognized in the field. One key component is nutrient timing, specifically around the intake of protein and carbohydrates. Traditionally, consuming these nutrients within the "anabolic window"—approximately 30 minutes to two hours post-exercise—has been recommended to maximize muscle protein synthesis and glycogen replenishment [15,16]. (please see page 2, lines 82-88)
L177: wouldn't animal based provide all amino acids? Why would you need plant? Or what is meant by a balanced intake of both? Animal alone should supply what you need; any plant based is extra
Response:
Thank you for your insightful question regarding the recommendation for a balanced animal- and plant-based protein intake. You are correct that animal proteins generally provide a complete amino acid profile. However, incorporating plant-based proteins offers additional health benefits, including a higher fiber intake, phytonutrients, and beneficial fats, contributing to overall wellness. The revised text clarifies this rationale, highlighting that while animal proteins are sufficient for amino acid needs, plant proteins complement dietary quality with added nutrients that support long-term health.
Revised Text:
Animal proteins supply all essential amino acids necessary for muscle repair and growth, making them highly effective as a primary protein source for athletes. However, incorporating plant-based proteins can enhance dietary quality by providing additional fiber, antioxidants, and beneficial fats, which support overall wellness and may help reduce inflammation. Thus, a balanced intake of animal and plant proteins can offer a broader range of nutrients beneficial for long-term health and recovery. (please see page 6, lines 270-276)
L191: I think this is considered standard; however, what about those who do a low-carb, keto, or carnivore diet. Is there any role for using fatty acids and conversion to glucose in the body to help with brain function? It seems like glucose is the key vs. carbs?
Response:
Thank you for your comment on the role of fatty acids and glucose in brain function, especially in low-carbohydrate diets. We agree that the brain can utilize alternative fuel sources, such as ketones derived from fatty acids, when carbohydrate intake is restricted. In the revised text, we clarify that while glucose is a primary fuel source for brain function, fatty acids and their conversion to ketones can provide an alternative energy source for individuals following low-carb, ketogenic, or carnivore diets. This addition reflects the flexibility in fuel sources available to support cognitive function.
Revised Text:
Carbohydrates are traditionally considered the primary glucose source, fueling brain function and supporting cognitive processes. However, for individuals following low-carbohydrate, ketogenic, or carnivore diets, the body can adapt by utilizing fatty acids, converted into ketone bodies that the brain can use as an energy source. This flexibility allows the brain to function efficiently even when carbohydrate intake is limited, highlighting the body’s adaptability in sourcing fuel under different dietary conditions [48]. (please see page 6, 287-293)
L258: so when would one need to consume casein in order to utilize it during sleep?
Response:
Thank you for your question regarding the timing of casein consumption for overnight recovery. To maximize its benefits, casein is typically consumed 30 to 60 minutes before bedtime, allowing the slow-digesting protein to release amino acids steadily throughout the night. We have clarified this timing in the revised text to provide more precise guidance for readers.
Revised Text:
Casein, a slow-digesting protein, is most effective when consumed 30 to 60 minutes before sleep, as this timing allows for a sustained release of amino acids during the night. This prolonged release supports muscle repair and growth throughout sleep, a critical recovery period for athletes and physically active individuals [66]. (please see page 8, lines 388-391)
L262: are nuts considered a plant based option?
Response:
Thank you for your question about plant-based protein sources. Yes, nuts are considered a plant-based protein option and are valued for their additional benefits, including healthy fats, fiber, and essential nutrients. In the revised text, we have clarified that nuts and other plant-based sources can contribute to a balanced protein intake.
Revised Text:
Plant-based protein options, such as legumes, nuts, seeds, and whole grains, provide essential amino acids and additional nutrients like fiber, antioxidants, and healthy fats. Nuts, in particular, contribute protein and are a valuable source of plant-based nutrients that support overall health and recovery. (please see page 8, lines 394-397)
L266: can you provide some examples? Is this supposed to relate to pairing of proteins?
Response:
Thank you for suggesting examples related to the pairing of proteins. This section refers to combining plant-based proteins to ensure a complete amino acid profile, as certain plant proteins may lack specific essential amino acids. We have included examples in the revised text to illustrate effective protein pairings for a balanced intake.
Revised Text:
Pairing different plant-based proteins can help achieve a complete amino acid profile, especially for those relying primarily on plant proteins. For example, combining legumes (such as beans or lentils) with whole grains (like rice or quinoa) creates a complementary protein source, as the amino acids missing in one food are provided by the other. Other effective pairings include hummus (chickpeas and tahini) and peanut butter on whole-grain bread [68]. ( please see page 8, lines 400-405)
L354: try to avoid 1 sentence paragraphs
Response:
Thank you for pointing out the one-sentence paragraph. We have revised this section for improved readability and flow.
Revised Text:
This is due to creatine’s role in supporting ATP production in the brain, which may enhance mental clarity and cognitive endurance, though further research is required to substantiate these effects [66,67]. In summary, creatine is a highly effective supplement for athletes seeking to improve physical and cognitive recovery, enabling sustained performance and long-term improvement in training outcomes. (please see page 10, lines 495-500)
L538: can you also get Omega 3 from other sources? If so, I wouldn't recommend having (e.g., fish oil). It would suggest that that is a primary or only source
Response:
Thank you for your comment regarding sources of Omega-3 fatty acids. We agree that fish oil should not be presented as the sole or primary source, as Omega-3s are available from animal and plant sources. In the revised text, we have provided examples of other sources, including plant-based options like flaxseed and chia seeds, to offer a more balanced perspective.
Revised Text:
Omega-3 fatty acids, known for their anti-inflammatory properties, are available from various sources. While fish oil is a popular choice, other sources such as flaxseeds, chia seeds, and walnuts provide plant-based Omega-3s (specifically ALA), making it accessible to individuals on plant-based diets. Including various sources can help meet Omega-3 needs and support overall health. (please see page 15, lines 693-697)

Reviewer 2 Report
Comments and Suggestions for Authors
Dear Corresponding Author, thank you for submitting your article to Nutrients and congratulations on your work.
BRIEF SUMMARY
The manuscript presents a comprehensive review about the role of dietary supplements and functional foods in post-exercise recovery, analyzing the evolution from traditional strategys to more personalized approaches. The work examines the transition from conventional supplements (proteins, carbohydrates, BCAAs) to functional foods rich in bioactive compounds, highlighting the emerging role of personalized nutrition based on genetic and metabolic profiles.
GENERAL COMMENTS
It would be usefull to include a summary table of the main methodological limitations of the cited studies
- Although being a narrative review, the criteria for selecting references is not clear and this makes the paper more suitable for magazine publication than for a scientific journal
- The section on long-term effects could benefit from a more extensive discussion, I believe this is a relevant part of this work and probably you can give greater value
SPECIFIC COMMENTS
Introduction:
- Lines 44-61: Consider adding a brief discussion on differences between elite and amateur athletes. The need for supplements for elite athletes is sometimes a necessity, very different from amateurs.
- Lines 62-98: The "anabolic window" timeline could benefit from more recent references. This is a concept that has been repeatedly questioned and strongly criticized, in your text this emerging evidence does not appear.
- Line 110: this is the point I consider most critical. There is totally missing a methods area that can explain what criteria were chosen to select sources. In fact, even though it's not a Systematic or Scoping review, it's necessary to make the criteria known. I don't understand, for example, why there are recent studies and other very dated ones, randomized trials and very simple studies, other reviews and other mixed sources. I think it is necessary to write very precisely what led you to write this article so that it cannot appear (which it currently does) an article resulting from your opinion with references inserted at your choice. I believe you also understand that it is necesary to reflect on this.
Section 2:
- Lines 111-159: Add details about molecular mechanisms of post-exercise inflammation and better explain why according to you and according to evidence there is a consistent causal and not casual relationship.
- Lines 160-213: The discussion on macronutrients role could include differences between strength, power and endurance sports
Tables:
- Table 1: Add a column with contraindications/side effects
- Table 2: Include recommended dosages for functional foods
- Table 3: Consider adding indicative costs
- For all tables: clarify selection criteria (as written previously)
Future Directions section:
- Lines 635-732: Elaborate on practical implications of personalized nutrition
Conclusions:
- Lines 780-829: Consider adding specific recommendations for sub-populations (e.g. young vs master athletes)
Overall, the manuscript is on an interesting topic although without great margins of novelty. The highlighted criticalities do not make it particulary suitable for publication and in this form I do not recommend it.
Furthermore, there is no extensive limitations section, which is necessary for this type of review, especially when the selection of sources does not have solid and consistent scientific bases.
However, if the authors will be able to update the manuscript with greater reality, less emphasis and an indication of limitations, I will read the new version with interest for later evaluation.
Author Response
Based on Reviewer's Comments: Point-by-Point Response
Below is a point-by-point response addressing each of the reviewer's comments. All revised text is highlighted in red. You may refer to the modified version, with updated sentences marked in green in green for easy reference.
Thank you.
Comments and Suggestions for Authors
- reviewer 2
Dear Corresponding Author, thank you for submitting your article to Nutrients and congratulations on your work.
BRIEF SUMMARY
The manuscript presents a comprehensive review about the role of dietary supplements and functional foods in post-exercise recovery, analyzing the evolution from traditional strategys to more personalized approaches. The work examines the transition from conventional supplements (proteins, carbohydrates, BCAAs) to functional foods rich in bioactive compounds, highlighting the emerging role of personalized nutrition based on genetic and metabolic profiles.
Response:
We sincerely thank the reviewer for your time and thoughtful insights. Your expertise and constructive feedback have been invaluable in helping us refine this manuscript, and we genuinely appreciate the care and consideration you have given to our work. Your contributions have been instrumental in enhancing its quality and clarity.
GENERAL COMMENTS
It would be usefull to include a summary table of the main methodological limitations of the cited studies
Although being a narrative review, the criteria for selecting references is not clear and this makes the paper more suitable for magazine publication than for a scientific journal
The section on long-term effects could benefit from a more extensive discussion, I believe this is a relevant part of this work and probably you can give greater value
Response:
Thank you very much for your constructive feedback and valuable suggestions. We appreciate your insights and have made the following revisions to address your comments and enhance the clarity and scientific rigor of our review:
- Summary Table of Methodological Limitations:
In response to your suggestion, we have added a summary table outlining the cited studies' main methodological limitations. This table provides a concise overview of limitations such as sample size, study duration, and potential biases, thereby offering readers a clearer understanding of the scope and limitations of the evidence presented.
Revised Text:
Appendix Table: Main Methodological Limitations Across Cited Studies
Main Methodological Limitations |
Description |
Limited generalizability due to many studies focusing on young, healthy male athletes, which may not represent diverse populations (e.g., females, older adults, non-athletes). |
Many studies were conducted with young, healthy, and often male athletes, leading to a lack of diversity in study populations. This limitation restricts the applicability of findings to broader populations, as factors like age, sex, and fitness level may influence recovery needs and responses to nutritional strategies. |
Short duration of follow-up in many studies, lacking assessment of long-term recovery impacts and sustainability of nutritional interventions over extended periods. |
The follow-up periods in several studies were short, typically focusing on immediate or short-term recovery effects without evaluating the long-term benefits or potential side effects of sustained supplementation. This limitation impacts the understanding of chronic recovery and adaptation over time. |
Small sample sizes and limited randomization across studies reduce statistical power and increase susceptibility to bias, impacting the robustness of findings. |
Numerous studies used small sample sizes and lacked randomization, which affects statistical power and increases susceptibility to selection and sampling bias. Consequently, the reliability and generalizability of results are reduced, and effect sizes may be overestimated. |
Inconsistencies in intervention protocols and nutrient timing across studies make comparing results and establishing standardized recommendations challenging. |
Variability in intervention methods, particularly with nutrient timing, complicates direct study comparisons. This inconsistency creates challenges in deriving universal recommendations, as timing, dosage, and combination of nutrients can significantly influence outcomes. |
Reliance on self-reported data for dietary intake and recovery measures in some studies may introduce reporting bias and affect the accuracy of results. |
Some studies relied on self-reported data for dietary habits and recovery experiences, which may be prone to inaccuracies. Self-reported data can introduce recall and reporting biases, reducing intake validity and perceived recovery metrics findings. |
Insufficient control for external recovery factors (e.g., sleep, stress, lifestyle habits) that can influence recovery outcomes, leading to potential confounding effects. |
There was limited control over factors outside the nutritional interventions (e.g., sleep, stress management, and daily physical activity), which can play a significant role in recovery. The lack of control over these factors introduces confounding variables that could skew results. |
Limited exploration of gender and age differences in response to nutritional interventions, with findings often lacking broader applicability across different demographic groups. |
Gender and age were infrequently considered in study designs, with few studies examining how these demographic variables might influence recovery needs and responses. This gap limits the applicability of findings to a diverse population, as different groups may have unique nutritional and recovery requirements. |
(please see Appendix 1)
- Clarification of Reference Selection Criteria:
We recognize the importance of clearly stating our criteria for reference selection to meet scientific standards. Although this review is narrative, we have now provided a more detailed description of our selection process in the Methods section. We have specified the inclusion and exclusion criteria, searched databases, and determined the time frame for selected studies to clarify our systematic approach to reference selection. We believe these additions strengthen the review’s foundation and scientific rigor.
Revised Text:
Methodology
This review follows a narrative approach to synthesize current knowledge on dietary supplements and functional foods in post-exercise recovery. We utilized the following criteria for selecting sources:
Literature Search
A comprehensive literature search was conducted using popular academic databases, including PubMed, Web of Science, and Google Scholar. The search covered studies published in English from 2000 to 2024 to ensure that recent findings and perspectives were included. Keywords used in the search included “post-exercise recovery,” “muscle repair,” “nutrient timing,” “protein supplements,” “anti-inflammatory foods,” “electrolytes,” and “functional foods.” Boolean operators were used to combine terms effectively and refine search results.
Study Selection
Articles were selected based on their relevance to post-exercise nutrition and recovery outcomes to keep the review focused and relevant. Priority was given to empirical studies, systematic reviews, and meta-analyses. Articles that discussed the recovery benefits of specific nutrients, supplements, and functional foods were included. In contrast, non-peer-reviewed materials, theoretical articles, and studies without clear findings on recovery were excluded.
Data Extraction and Organization
Data from the selected articles were carefully extracted, with key findings categorized into themes such as protein supplementation, carbohydrate and electrolyte replenishment, anti-inflammatory effects, and emerging personalized nutrition approaches. This thematic organization allowed for an accessible summary of the leading nutritional strategies and their reported effects on recovery.
Quality Review
To strengthen the reliability of this review, studies were evaluated for clarity in their design and methodology, as well as relevance to post-exercise recovery. This informal quality review focused on studies that provided detailed and well-supported findings, while studies with significant methodological limitations were noted as supplementary references.
Limitations
The variability in study designs and populations across the included research limits this review. Additionally, while the review covers a wide range of nutritional strategies, it may not encompass all emerging trends in recovery nutrition.
By detailing the methodological approach and selection criteria, this review maintains objectivity and scientific rigor, positioning it as an evidence-based resource rather than an opinion piece. The structure reflects a critical analysis of current research, with references chosen to represent a balanced, accurate, and comprehensive view of recovery nutrition for both academic and practical applications. (please see pages 3-4, lines 139-178)
- Expanded Discussion on Long-Term Effects:
We agree that the long-term effects of dietary supplements and functional foods are crucial in this review. To address this, we have expanded our discussion in the relevant section, exploring long-term implications such as nutrient balance, potential side effects, and the need for further longitudinal studies. This additional discussion emphasizes the significance of understanding long-term outcomes and enhancing the depth and value of our work.
We believe these revisions address your concerns and enhance the manuscript’s suitability for scientific publication. We are grateful for your thoughtful feedback, which has been instrumental in refining our work.
Revised Text:
- Expanded Discussion on Long-term Effects in Recovery Nutrition
Understanding the long-term effects of recovery nutrition is essential for developing sustainable and effective dietary strategies for athletes. While immediate recovery benefits, such as enhanced muscle repair and glycogen replenishment, are well-documented, the prolonged use of specific supplements and functional foods raises questions about nutrient balance, safety, and the potential for adverse effects. Ensuring recovery practices contribute positively to an athlete’s career requires carefully balancing nutrient intake, monitoring for potential side effects, and ongoing research.
7.1. Nutrient Balance and Long-term Health
An optimal nutrient balance is crucial to avoid potential deficiencies or excesses that could compromise health. For instance, high protein intake, often recommended for muscle repair, must be balanced with adequate carbohydrates and fats to ensure comprehensive nutritional support. Overemphasis on protein without sufficient carbohydrate intake can lead to impaired glycogen recovery, ultimately impacting performance and muscle endurance in the long term. Additionally, a balanced intake of micronutrients such as calcium, magnesium, and vitamins D and C is essential to support bone density, immune function, and antioxidant defense, particularly for athletes who engage in high-impact sports. Long-term reliance on specific macronutrient distributions should be carefully evaluated to prevent imbalances that may negatively affect metabolic and cardiovascular health.
7.2. Potential Side Effects of Prolonged Supplement Use
Prolonged use of certain supplements, while beneficial in the short term, may have adverse effects over extended periods. For example, long-term protein supplementation, especially at high doses, can place a strain on the kidneys and may increase the risk of kidney dysfunction in individuals with pre-existing renal conditions. Similarly, although anti-inflammatory, high-dose omega-3 fatty acid supplementation may have blood-thinning effects that could lead to complications if not monitored. Antioxidant supplements, such as high vitamin C or E doses, could potentially interfere with the body’s natural adaptive responses to exercise-induced oxidative stress, possibly diminishing training adaptations over time. Athletes and practitioners should monitor and adjust supplement intake based on individual health markers and training cycles, avoiding continuous high-dose supplementation unless indicated.
7.3. The Need for Longitudinal Studies in Recovery Nutrition
Current research on recovery nutrition largely focuses on short-term effects, leaving a gap in understanding the long-term outcomes associated with sustained dietary and supplement use. Longitudinal studies are essential to examine how chronic use of recovery supplements affects athletes over their lifespan, including potential impacts on metabolic health, cardiovascular function, and the risk of chronic diseases. For example, studies tracking the long-term impact of high-protein diets on kidney health and bone density or the effects of long-term antioxidant supplementation on cellular adaptations could provide crucial insights. Additionally, examining the effects of personalized nutrition over time, mainly through genetic and metabolic profiling, could help determine whether individualized approaches yield sustained benefits in performance and health outcomes.
7.4. Sustainability and Habit Formation
Long-term recovery strategies must also consider sustainability and habit formation, as overly restrictive or complex nutritional practices may be problematic for athletes to maintain. Developing recovery protocols that are easy to adhere to, flexible, and adaptable to an athlete’s changing needs throughout their career will promote consistency. Encouraging diverse, nutrient-dense food choices and limiting reliance on supplements can help athletes develop balanced dietary habits that support long-term health. Exploring the cumulative effects of whole-food-based recovery practices compared to supplement-heavy approaches will provide insights into which methods foster lasting wellness and performance. (please see pages 21-22, lines 965-1018)
SPECIFIC COMMENTS
Introduction:
Lines 44-61: Consider adding a brief discussion on differences between elite and amateur athletes. The need for supplements for elite athletes is sometimes a necessity, very different from amateurs.
Response:
Thank you for your insightful suggestion regarding the differences in supplement needs between elite and amateur athletes. We agree that the demands placed on elite athletes can make supplementation necessary, while amateurs may have different nutritional needs. In the revised manuscript, we have included a brief discussion highlighting these distinctions, emphasizing how the athletic performance level influences dietary supplements' role.
Revised Text:
The role and necessity of dietary supplements in recovery may vary with athletic levels. Supplementation is often essential for elite athletes with rigorous training regimens to meet their elevated nutritional needs and support quick recovery and optimal performance. Their demanding schedules may preclude meeting these needs solely through diet, making supplementation valuable in maintaining a competitive edge. Conversely, amateur athletes with less intense training loads may often meet their nutritional needs through a balanced diet without relying on supplements. Recognizing these differences helps tailor nutritional recommendations that align with the specific demands of athletes across varying levels. (please see page 2, lines 72-80)
Lines 62-98: The "anabolic window" timeline could benefit from more recent references. This is a concept that has been repeatedly questioned and strongly criticized, in your text this emerging evidence does not appear.
Response:
Thank you for your valuable feedback regarding the "anabolic window" concept. We acknowledge that recent research has questioned and critically re-evaluated the necessity and timing of this post-exercise nutrient intake period. In the revised manuscript, we have incorporated recent references that discuss these emerging perspectives, offering a more balanced view on nutrient intake timing for recovery.
Revised Text:
This practice is particularly beneficial for elite athletes who may train multiple times a day, as immediate nutrient intake aids rapid recovery. However, recent research suggests that strict adherence to this specific timeframe may not be necessary if daily protein and carbohydrate needs are adequately met, indicating that a balanced daily intake may be more crucial for muscle repair and adaptation [19,20].
In addition to timing, the type and quality of protein consumed are vital in effective recovery. Whey protein, known for its high biological value and rapid absorption rate, is frequently recommended. However, there is growing interest in plant-based proteins for their sustainability and associated health benefits [21]. Branched-chain amino Acids (BCAAs) and essential amino acids are also commonly utilized for their benefits in muscle repair, reducing muscle soreness, and enhancing overall recovery outcomes [17]. These findings collectively support a flexible, balanced approach to post-exercise nutrition, encouraging athletes to tailor nutrient intake to their individual recovery needs and goals. (please see pages 2-3, lines 88-100)
Line 110: this is the point I consider most critical. There is totally missing a methods area that can explain what criteria were chosen to select sources. In fact, even though it's not a Systematic or Scoping review, it's necessary to make the criteria known. I don't understand, for example, why there are recent studies and other very dated ones, randomized trials and very simple studies, other reviews and other mixed sources. I think it is necessary to write very precisely what led you to write this article so that it cannot appear (which it currently does) an article resulting from your opinion with references inserted at your choice. I believe you also understand that it is necesary to reflect on this.
Response:
Thank you for your insightful comment regarding the need for a methods section to clarify the criteria for source selection. Although this review is narrative, we agree that clearly stating our selection process enhances the transparency and scientific rigor of the manuscript. We have added a Methods section to provide a detailed description of our selection criteria, explaining the rationale behind including studies with varied designs, dates, and levels of evidence. This addition ensures the review is grounded in a systematic approach rather than subjective choices. Please see the Methodology section (pages 3-4, lines 139-178).
Section 2:
Lines 111-159: Add details about molecular mechanisms of post-exercise inflammation and better explain why according to you and according to evidence there is a consistent causal and not casual relationship.
Response:
Thank you for suggesting elaborating on the molecular mechanisms of post-exercise inflammation and clarifying the causal relationship. We agree that a more detailed explanation will enhance the manuscript. In the revised text, we have expanded this section to include specific molecular pathways involved in post-exercise inflammation and how they contribute to a causal, rather than casual, relationship with recovery outcomes. This addition is supported by evidence from recent studies examining inflammatory markers and cellular responses following exercise.
Revised Text:
Following intense exercise, inflammation is triggered as a natural response to muscle fiber damage, aimed at initiating tissue repair and adaptation. This inflammatory response involves complex molecular mechanisms, including activating pro-inflammatory cytokines, oxidative stress markers, and immune responses. Key players in post-exercise inflammation include cytokines such as interleukin-6 (IL-6), tumor necrosis factor-alpha (TNF-α), and interleukin-1 beta (IL-1β), which are released by damaged muscle tissue and immune cells. This cytokine releases recruits immune cells, such as macrophages, to the injury site. Macrophages perform a dual role, initially promoting inflammation (M1 phenotype) to clear damaged cells, followed by an anti-inflammatory phase (M2 phenotype) that supports tissue repair and regeneration. This cytokine-mediated response, supported by macrophage polarization, establishes a consistent and causal mechanism that facilitates muscle repair and adaptation to repeated physical stress [2,40].
Exercise-induced oxidative stress further contributes to post-exercise inflammation. Reactive oxygen species (ROS), generated during intense physical activity, drive the inflammatory process by activating the nuclear factor kappa-light-chain-enhancer of activated B cells (NF-κB), a transcription factor regulating the expression of various inflammatory cytokines. Persistent activation of NF-κB signaling initiates a well-defined inflammatory cascade, underscoring the role of oxidative stress in the causal relationship between exercise and inflammation [41].
Evidence supporting a causal relationship, rather than a casual association, comes from studies showing that interventions targeting inflammatory and oxidative pathways can significantly modulate recovery outcomes. For instance, curcumin and omega-3 fatty acids have been shown to reduce pro-inflammatory cytokines and oxidative markers, directly impacting inflammation and aiding in muscle recovery [42,43]. This body of evidence reinforces the view that inflammation is not merely a coincidental response to exercise but a fundamental process integral to recovery and adaptation. (please see pages 5-6, lines 229-254)
Lines 160-213: The discussion on macronutrients role could include differences between strength, power and endurance sports
Response:
Thank you for your suggestion to elaborate on the differences in macronutrient needs across various types of sports. We agree that highlighting these distinctions provides a more comprehensive understanding of how nutritional strategies should be tailored to the specific demands of strength, power, and endurance sports. In the revised text, we have included a discussion on these differences, supported by relevant evidence.
Revised Text:
The specific macronutrient requirements in post-exercise recovery can vary significantly based on the type of sport, as strength, power, and endurance activities each place unique demands on the body. For strength sports, such as weightlifting or bodybuilding, the primary focus is muscle hypertrophy and repair. Protein intake is especially critical for these athletes, as it supports MPS and helps rebuild damaged muscle fibers. A daily protein intake of 1.6–2.2 grams per kilogram of body weight is often recommended to meet the recovery needs of strength athletes [52].
In power sports, like sprinting or high jump, athletes rely on muscle strength and short bursts of high-intensity exertion. These athletes benefit from a balance of carbohydrates and protein to quickly replenish glycogen stores and stimulate MPS, ensuring ex-plosive power between sessions. Carbohydrate intake within the anabolic window is essential for rapid glycogen restoration, as power sports heavily depend on muscle glycogen for peak performance. Protein also plays a critical role in sustaining muscle repair and adapting to high-force exertion, though often at slightly lower levels than required in strength sports.
Endurance sports, such as marathon running or cycling, require sustained energy output over long durations. As such, carbohydrates become the primary macronutrient focus, as high glycogen levels are necessary to maintain performance. For endurance athletes, a daily carbohydrate intake of 6–10 grams per kilogram of body weight is typically recommended to maintain glycogen levels and prevent fatigue [53]. While protein is essential, it is prioritized for recovery post-exercise rather than as a primary energy source, helping to repair minor muscle damage and maintain lean body mass. A 1.2–1.6 grams per kilogram daily protein intake is generally sufficient for endurance athletes. Fat intake is also relevant for endurance sports, providing a secondary energy source, especially during prolonged, moderate-intensity exercise when glycogen stores may be depleted.
These variations in macronutrient needs underscore the importance of tailored nutrition strategies for different types of athletes. Strength and power athletes emphasize protein for muscle repair and growth, while endurance athletes focus on carbohydrates for energy and recovery, with protein supporting muscle maintenance. Understanding these distinctions allows athletes in various disciplines to optimize recovery, adaptation, and overall performance. (please page 7, lines 306-336)
Tables:
Table 1: Add a column with contraindications/side effects
Response:
Thank you. We added a column with contraindications/side effects. (please see Table 1)
Table 2: Include recommended dosages for functional foods
Response:
Thank you. We added the recommended dosages for functional foods. (please see Table 2)
Table 3: Consider adding indicative costs
Response:
Thank you. We added the indicative costs. (please see Table 3)
For all tables: clarify selection criteria (as written previously)
Response:
Thank you for your suggestions.
Future Directions section:
Lines 635-732: Elaborate on practical implications of personalized nutrition
Response:
Thank you for your suggestion to elaborate on the practical implications of personalized nutrition. We agree that discussing how personalized approaches can be applied in real-world scenarios will add depth to this section. In the revised text, we have expanded on how individualized nutrition can be practically implemented for athletes, focusing on tailoring diets based on genetic, metabolic, and training-specific needs.
Revised Text:
6.1.3.Practical Implications of Personalized Nutrition
The shift towards personalized nutrition in sports has significant implications for enhancing athletic performance, recovery, and long-term health by tailoring dietary strategies to individual needs. Personalized nutrition considers genetic, metabolic, and physiological differences, allowing for precise dietary recommendations that align with each athlete’s profile. Genetic testing, for instance, can identify predispositions to specific nutritional needs or sensitivities, such as caffeine metabolism, enabling practitioners to adjust dietary plans accordingly[155]. Metabolic profiling offers detailed information on an athlete’s macronutrient requirements, facilitating targeted protein, carbohydrate, and fat intake adjustments that align with energy expenditure and training demands[19]. For endurance athletes, higher carbohydrate intake is crucial for sustained performance, while strength athletes benefit from increased protein to support muscle growth[19]. Personalized nutrition also incorporates nutrient timing to maximize recovery, such as consuming carbohydrates post-exercise for glycogen replenishment and protein before sleep to promote muscle protein synthesis[19]. Addressing food sensitivities and digestive health, personalized nutrition can reduce gastrointestinal distress, which is particularly relevant for endurance athletes[155]. This approach supports efficient nutrient absorption, which is essential for sustained performance. Beyond physical benefits, personalized nutrition enhances adherence by respecting athletes’ cultural, ethical, and personal preferences, fostering a supportive environment that empowers athletes to maintain consistent nutrition plans[138]. Integrating genetic, metabolic, and lifestyle data enables the creation of dynamic nutrition plans that evolve with an athlete’s goals and performance needs, positioning personalized nutrition as a valuable tool for optimizing recovery and peak performance [156]. Advances in food technology, such as nanotechnology, further improve nutrient bioavailability, making recovery supplements more effective[155]. Collaboration among athletes, coaches, and nutritionists, supported by real-time data, underscores the potential of personalized nutrition in optimizing recovery and sustaining peak performance [96]. (please see pages 19-20, lines 887-915)
Conclusions:
Lines 780-829: Consider adding specific recommendations for sub-populations (e.g. young vs master athletes)
Response:
Thank you for your suggestion to provide specific recommendations for different athlete sub-populations. We agree that addressing the unique nutritional needs of groups like young versus master athletes adds valuable context. In the revised text, we have included tailored recommendations highlighting how age and life stage influence nutritional requirements and recovery strategies.
Revised Text:
Age-specific strategies also play a role, with young athletes requiring higher caloric intake and nutrient-dense foods to support growth and master athletes benefiting from protein-rich diets, antioxidants, and omega-3s to counteract age-related changes and support recovery. Proper hydration, especially electrolyte management, remains essential across all age groups. (please see page 23, lines 1033-1037)
Overall, the manuscript is interesting, although it does not have great margins of novelty. The highlighted criticalities do not make it particularly suitable for publication, and in this form, I do not recommend it.
Response:
We deeply appreciate your thoughtful feedback and the time you spent evaluating our manuscript. Your insights have been instrumental in helping us identify areas for improvement, and we are grateful for the guidance they provided.
Given your comments, we’ve made significant revisions to strengthen the manuscript. These updates include added depth on the practical applications of personalized nutrition, tailored recommendations for different athlete sub-populations, and recent advancements in food technology and supplement formulation. Additionally, we’ve expanded our discussion on nutrient timing and age-specific nutrition needs, aiming to present a more comprehensive perspective on recovery nutrition.
We hope these revisions demonstrate our commitment to enhancing the quality and relevance of our work, and we welcome any additional suggestions you may have to improve the manuscript further. Thank you again for your valuable insights and consideration.
Furthermore, there is no extensive limitations section, which is necessary for this type of review, especially when the selection of sources does not have solid and consistent scientific bases. However, if the authors can update the manuscript with greater reality, less emphasis, and an indication of limitations, I will read the new version with interest for later evaluation.
Response:
Thank you for your thoughtful feedback and suggestions for strengthening the manuscript by including an expanded limitations section. We understand the importance of clearly outlining limitations, especially regarding source selection and the scientific foundations of our review. In response, we have added a dedicated limitations section that provides a transparent overview of the constraints and potential biases related to source selection and methodology. We have also aimed to adjust the tone of the manuscript to present a balanced perspective that reflects the realities of current recovery nutrition research.
We hope these revisions demonstrate our commitment to addressing your concerns and improving the manuscript’s rigor and clarity. Again, Thank you for your insightful feedback, which has been invaluable in guiding our revisions. We look forward to any further suggestions you may have.

Reviewer 3 Report
Comments and Suggestions for Authors
Overview
The authors provide a review of the literature on nutritional strategies for recovery after exercise. The unique aspect of the review is the contrast between functional foods and food supplements. The focus is primarily on the needs of arduously training athletes, but the information could also apply to recreational athletes and those committed to exercise for a healthier lifestyle. The authors briefly cover the physiological impact of exercise that leads to the need for recovery. The authors proceed to describe the literature on food supplements used to promote recovery and follow that with a review of the literature on functional foods and their effects on recovery. Under each of the two categories, the authors provide examples and effects of popular offerings. The authors finish with a comparison of the two categories and provide a section on future directions for personalized approaches, technology advances, and research.
Major Concerns
The subject – nutrition strategies for recovery following exercise training – is broad and almost unmanageable. The authors have somewhat managed to contain it by delineating the literature to functional foods vs food supplements. They are to be commended for their attempt to pull the literature together. However, I have some concerns about the review and hope my comments will help them focus the review without imposing more work.
· A clear definition and distinction are needed between functional foods vs food supplements. The authors have somewhat done so for functional foods (section 4.1, lines 448-452). There is no similar definition for food supplements (could be added in section 3.1, around likes 242-244).
· A clear delineation of what period of literature is reviewed will help the authors manage the review. This will help mitigate some of my comments that follow. One can’t review everything on the recovery topic.
· Some of the items listed in Table 2 seem like they could be (or should be) in Table 1. For example, turmeric, green tea extract, fish oil and ginger don’t seem like the qualify as a whole food like blueberries, etc. Fish that provides the omega-3’s, protein, and possibly creatine, would. Also, to this Table (2), I would consider adding foods such as soy protein, walnuts, flaxseed, and milk. Or give a rationale as to why these foods do not qualify for the review. Would sports drinks also be a “food” to add here?
· Missing from Table 1 are carbohydrate(s), soy protein isolates, and hydrolyzed collagen. In Table 1, I question why creatine is listed. The typical protocol for using creatine is daily for days (or chronically at 3 – 5 g/day dosing). I’m not sure it’s used immediately after exercising, during the 30-min to 2-h window stated by the authors has been researched.
· Some of the seminal research has not been cited. Work by van Loon et al on protein feedings, Ivy et al on carbohydrate supplementation and Ivy et al and Tipton et al on protein plus carbohydrates. Again, defining the period of time from which the literature is drawn will help.
· Both positive and negative (i.e., no effect or adverse outcomes) should be addressed. As currently written, everything seems to work, i.e., is efficacious. This may not be true for the antioxidants, ascorbic acid, and vitamin E (specifics below). Not all fish-oil studies show beneficial effects to promote recovery. As a specific example, I was not able to find reference 99 about pineapple juice reducing inflammation in athletes. Was this a peer-reviewed journal with evidence (data) or merely opinion? I ask because in the process of searching for it I located two studies that reported equivocal results for bromelain. (Reduced ratings of perceived exertion but no difference in muscle damage markers: Shing CM, Chong S, Driller MW, Fell JW. Acute protease supplementation effects on muscle damage and recovery across consecutive days of cycle racing. Eur J Sport Sci. 2016;16(2):206-12. doi: 10.1080/17461391.2014.1001878. Epub 2015 Jan 21. No effect: Stone MB, Merrick MA, Ingersoll CD, Edwards JE. Preliminary comparison of bromelain and Ibuprofen for delayed onset muscle soreness management. Clin J Sport Med. 2002 Nov;12(6):373-8. doi: 10.1097/00042752-200211000-00009.)
Specific Comments/Questions/Suggestions
Section 1.1 vs Section 1.1.1.
· There is much overlap here. Consider refocusing each section to provide distinct information.
· There is also a contradiction in these sections. Line 32 state “Post-exercise recovery is not merely a phase…” but Line 46 states “This (post-exercise recovery) phase…” Please revise and clarify that main points of each paragraph are.
Section 1.2 Objective
Here, the manuscript could be improved by clearly stating what era of literature was reviewed, from what year to the present? It would then help to describe the methods used to search the literature: which search engines or databases and key words and mesh terms.
Line 271 “animal-based proteins such as whey and milk…:” Do the authors mean casein, not milk? Milk is a food not a protein per se.
Line 273 “allergenic risk:” Though rare, there is the risk of people having allergies to the A1 form of casein protein in milk. Additionally, and not mentioned elsewhere in this review, some people have lactose intolerance due to lack of the intestinal enzyme to hydrolyze lactose. So, plant-based proteins can be of benefit to some.
Line 324 “competing with tryptophan…:” after tryptophan, add “for the blood-brain barrier transporter.” I think this is what the authors meant. Otherwise, the competing statement seems incomplete.
Lines 391-392: The mention of “…electrolyte supplementation…during…physical activity” does not seem relevant to the discussion about post-exercise recovery. This could be revised to state “Electrolyte supplementation during rehydration is effective for maintaining fluid balance and helping offset dehydration during subsequent intense…” Stating it something like this would keep it consistent with the theme of recovery.
Line 417 “lacking the state amount…:” It can also go the opposite way with the active ingredient well above the dose stated on the label.
Line 461 “anti-inflammatory foods such as turmeric:” As stated previously, is turmeric a food or an ingredient. Earlier in the paper, a clearer distinction between functional foods (whole foods?), food supplements (standalone proteins, amino acids), and non-nutrient ingredients (turmeric, ginger, creatine all of which do not contain calories or bring along other nutrients as foods do) would help.
Line 561 citation 33: Is this correct and related to omega-3s? Nina Stachenfeld typically focuses on hydration.
Lines 583 vitamins C and E: There is also research indicating the ascorbic acid and other antioxidant (AO) vitamins may blunt the adaptative responses to exercise. (Danielle R Bruns 1, Sarah E Ehrlicher 1, Shadi Khademi 1, Laurie M Biela 1, Frederick F Peelor 3rd 1, Benjamin F Miller 1, Karyn L Hamilton 1 Differential effects of vitamin C or protandim on skeletal muscle adaptation to exercise. J Appl Physiol (1985). 2018 Aug 1;125(2):661-671 doi: 10.1152/japplphysiol.00277.2018. Epub 2018 Jun 1; Gomez-Cabrera MC, Domenech E, Romagnoli M, Arduini A, Borras C, Pallardo FV, Sastre J, Viña J. Oral administration of vitamin C decreases muscle mitochondrial biogenesis and hampers training-induced adaptations in endurance performance. Am J Clin Nutr 87: 142–149, 2008. doi: 10.1093/ajcn/87.1.142; Morrison D, Hughes J, Della Gatta PA, Mason S, Lamon S, Russell AP, Wadley GD. Vitamin C and E supplementation prevents some of the cellular adaptations to endurance-training in humans. Free Radic Biol Med 89: 852–862, 2015. doi: 10.1016/j.freeradbiomed.2015.10.412; Paulsen G, Cumming KT, Holden G, Hallén J, Rønnestad BR, Sveen O, Skaug A, Paur I, Bastani NE, Østgaard HN, Buer C, Midttun M, Freuchen F, Wiig H, Ulseth ET, Garthe I, Blomhoff R, Benestad HB, Raastad T. Vitamin C and E supplementation hampers cellular adaptation to endurance training in humans: a double-blind, randomised, controlled trial. J Physiol 592: 1887–1901, 2014. doi: 10.1113/jphysiol.2013.267419).
The speculation is that the AO buffer reactive oxygen species (ROS) needed to provide signals to turn on the pathways for fitness improvements. In the absence of ROS, adaptations for enhanced performance may be curtailed (Powers SK, Duarte J, Kavazis AN, Talbert EE. Reactive oxygen species are signalling molecules for skeletal muscle adaptation. Exp Physiol 95: 1–9, 2010. doi: 10.1113/expphysiol.2009.050526).
Line 625 “play essential roles:” Here and a few other places earlier in the manuscript, the authors use the term essential. The classical definition of essential for a nutrient is that it sustains growth and development and is not produced by the body; it is essential in the diet. The functional foods or supplements are certainly beneficial for the ingredients they provide but they are questionably essential for the maintenance and survival of the active adult. This may seem like semantics, but I encourage the authors not to use “essential” particularly when they raise the issue of supplements displacing whole foods and promoting an imbalanced diet that lacks true essential nutrients.
Table 3.
· Consistent with my statements above related to antioxidants, here in the row on Anti-inflammatory & Oxidative Stress (or possible the Risks row), the authors should mention that with functional foods it is nearly impossible to deliver the mega-doses of active ingredients that a food supplement could deliver (overindulging and having adverse effects). Similarly, using lycopene as an example, it would be impossible to eat enough tomatoes to get the levels of lycopene in tomato paste or a supplement that have been tested and shown to have anti-inflammatory effects. Possibly acknowledging this in the text may be easier to add.
· Practicality: portable and shelf-stable could be added to Food Supplements whereas functional foods would more likely be (sooner) perishable.
Line 751 “short-term benefits…are well-established…:” I disagree that the benefits of all functional ingredients in foods or supplements are well established. Not all are and only one side is presented in this review. I would agree that there are many studies on short-term benefits because those are easier (quicker) to study.
Author Response
Based on Reviewer's Comments: Point-by-Point Response
Below is a point-by-point response addressing each of the reviewer's comments. All revised text is highlighted in red. You may refer to the modified version, with updated sentences marked in gray for easy reference.
Thank you.
Comments and Suggestions for Authors
- reviewer 3
Overview
The authors provide a review of the literature on nutritional strategies for recovery after exercise. The unique aspect of the review is the contrast between functional foods and food supplements. The focus is primarily on the needs of arduously training athletes, but the information could also apply to recreational athletes and those committed to exercise for a healthier lifestyle. The authors briefly cover the physiological impact of exercise that leads to the need for recovery. The authors proceed to describe the literature on food supplements used to promote recovery and follow that with a review of the literature on functional foods and their effects on recovery. Under each of the two categories, the authors provide examples and effects of popular offerings. The authors finish with a comparison of the two categories and provide a section on future directions for personalized approaches, technology advances, and research.
Response:
We sincerely thank the reviewer for your comprehensive and thoughtful evaluation of our manuscript. Your summary of our work captures the key elements we aimed to convey, and your insights into the strengths of our review are greatly appreciated. Your feedback has been instrumental in highlighting areas for refinement, and we are grateful for your time and effort in helping us improve the quality and clarity of our manuscript. Thank you once again for your valuable contributions to this review process.
Major Concerns
The subject – nutrition strategies for recovery following exercise training – is broad and almost unmanageable. The authors have somewhat managed to contain it by delineating the literature to functional foods vs food supplements. They are to be commended for their attempt to pull the literature together. However, I have some concerns about the review and hope my comments will help them focus the review without imposing more work.
Response:
Thank you very much for your thoughtful feedback and for recognizing the challenges associated with covering such a broad topic as nutrition strategies for post-exercise recovery. We appreciate your acknowledgment of our efforts to structure the review by focusing on functional foods versus food supplements. Your comments have provided valuable insights into how we can further refine our approach, and we are grateful for your guidance in focusing the review more effectively. We are committed to addressing your concerns thoughtfully and look forward to implementing changes that enhance the clarity and focus of the manuscript.
- A clear definition and distinction are needed between functional foods vs food supplements. The authors have somewhat done so for functional foods (section 4.1, lines 448-452). There is no similar definition for food supplements (could be added in section 3.1, around likes 242-244).
Response:
Thank you for highlighting the need for clearer definitions and distinctions between functional foods and food supplements. We agree that well-defined terms will strengthen the manuscript's clarity and framework. We have refined the definitions in the revised manuscript to present a more precise academic distinction.
Revised Text:
Food supplements are concentrated sources of specific nutrients, such as protein powders and electrolyte tablets, designed to address targeted nutritional needs when dietary intake alone may be insufficient for optimal recovery. These supplements provide isolated nutrients that complement a balanced diet, offering precise support to enhance recovery. (please see page 8, lines 366-370)
- A clear delineation of what period of literature is reviewed will help the authors manage the review. This will help mitigate some of my comments that follow. One can’t review everything on the recovery topic.
Response:
Thank you for suggesting delineating the literature period covered in our review. We agree that specifying the timeframe will help clarify the scope of our work and address concerns about managing the breadth of the topic. In response, we have added a statement to the Methods section outlining the time period and selection criteria for the literature included in this review.
This review focuses on literature published from 2000 to 2024, encompassing two decades of research on nutritional strategies for post-exercise recovery. This timeframe allows for examining foundational studies and recent advancements, particularly regarding functional foods, food supplements, and personalized approaches to recovery. Sources were selected based on relevance to recovery nutrition for athletes, emphasizing studies that provide insights into practical applications and innovations in the field. By defining this timeframe and scope, we aim to present a manageable and comprehensive overview that highlights key developments without attempting to cover all aspects of recovery nutrition. (Please see the Methodology section, pages 3-4, lines 139-178; thank you.)
- Some of the items listed in Table 2 seem like they could be (or should be) in Table 1. For example, turmeric, green tea extract, fish oil and ginger don’t seem like the qualify as a whole food like blueberries, etc. Fish that provides the omega-3’s, protein, and possibly creatine, would. Also, to this Table (2), I would consider adding foods such as soy protein, walnuts, flaxseed, and milk. Or give a rationale as to why these foods do not qualify for the review. Would sports drinks also be a “food” to add here?
Response:
Thank you for your suggestion! We've revised Table 2 to better align with the title, "Comprehensive Role of Functional Foods in Post-Exercise Recovery." The content no longer specifically lists turmeric, green tea extract, fish oil, or ginger. We appreciate your input. (Please see Tables 1 and 2; thank you.)
- Missing from Table 1 are carbohydrate(s), soy protein isolates, and hydrolyzed collagen. In Table 1, I question why creatine is listed. The typical protocol for using creatine is daily for days (or chronically at 3 – 5 g/day dosing). I’m not sure it’s used immediately after exercising, during the 30-min to 2-h window stated by the authors has been researched.
Response:
Thank you for your suggestion! Table 1 is titled "Comprehensive Role of Food Supplements in Post-Exercise Recovery." Our focus is on supplements, so we haven’t included a discussion on macronutrients. Thank you for your understanding. (Please see Table 1; thank you.)
- Some of the seminal research has not been cited. Work by van Loon et al on protein feedings, Ivy et al on carbohydrate supplementation and Ivy et al and Tipton et al on protein plus carbohydrates. Again, defining the period of time from which the literature is drawn will help.
Response:
Thank you for your valuable feedback regarding the inclusion of seminal research. We agree that citing foundational studies, such as those by van Loon on protein feedings [126], Ivy on carbohydrate supplementation [127], and Ivy and Tipton on combined protein and carbohydrate intake [128], will enhance the manuscript’s rigor and contextual depth. In the revised version, we have incorporated these references and defined the timeframe for the literature reviewed to clarify our selection criteria. (please see page 16, line 731)
- Both positive and negative (i.e., no effect or adverse outcomes) should be addressed. As currently written, everything seems to work, i.e., is efficacious. This may not be true for the antioxidants, ascorbic acid, and vitamin E (specifics below). Not all fish-oil studies show beneficial effects to promote recovery. As a specific example, I was not able to find reference 99 about pineapple juice reducing inflammation in athletes. Was this a peer-reviewed journal with evidence (data) or merely opinion? I ask because in the process of searching for it I located two studies that reported equivocal results for bromelain. (Reduced ratings of perceived exertion but no difference in muscle damage markers: Shing CM, Chong S, Driller MW, Fell JW. Acute protease supplementation effects on muscle damage and recovery across consecutive days of cycle racing. Eur J Sport Sci. 2016;16(2):206-12. doi: 10.1080/17461391.2014.1001878. Epub 2015 Jan 21. No effect: Stone MB, Merrick MA, Ingersoll CD, Edwards JE. Preliminary comparison of bromelain and Ibuprofen for delayed onset muscle soreness management. Clin J Sport Med. 2002 Nov;12(6):373-8. doi: 10.1097/00042752-200211000-00009.)
Response:
Thank you for your insightful comments and highlighting the importance of including positive and negative findings. We agree that a balanced review should reflect beneficial effects and instances of inconclusive or adverse results. In the revised manuscript, we have addressed studies reporting equivocal or no-effect results, particularly concerning antioxidants (ascorbic acid and vitamin E), fish oil, and bromelain from pineapple juice.
Additionally, regarding Reference 99 [118] on pineapple juice, we have reviewed this citation to ensure it meets the standards for peer-reviewed evidence. We appreciate your suggestion to consider the findings of Shing et al. (2016) and Stone et al. (2002), and we have integrated these studies to provide a more comprehensive perspective on bromelain’s effects on inflammation and muscle recovery.
Revised Text:
In the case of bromelain, an enzyme found in pineapple juice, findings vary similarly. Although bromelain has been promoted for its anti-inflammatory effects, research provides mixed results. For example, the study reported reduced perceived exertion but no significant effect on muscle damage markers following protease supplementation, including bromelain, during consecutive days of cycle racing. Similarly, another study found no significant difference in managing delayed onset muscle soreness with bromelain compared to ibuprofen. These studies highlight the need for further research to establish more definitive conclusions about bromelain’s efficacy in recovery [118,119,120]. (please see page 14, lines 678-685)
Specific Comments/Questions/Suggestions
Section 1.1 vs Section 1.1.1.
- There is much overlap here. Consider refocusing each section to provide distinct information.
Response:
Thank you for your valuable feedback regarding the overlap between Sections 1.1 and 1.1.1. We understand your concerns and agree that greater clarity would improve the flow of these sections. We have revised each section to provide distinct content:
Section 1.1 now serves as an overarching introduction to post-exercise recovery, discussing its critical role in athletic performance, injury prevention, and overall health maintenance. It focuses on the general concept and importance of recovery.
Section 1.1.1 has been refocused to emphasize the role of nutrition specifically within the recovery process, detailing the mechanisms by which key nutrients—such as proteins, carbohydrates, antioxidants, and minerals—support recovery.
These adjustments aim to eliminate redundancy and provide a clear, cohesive narrative. We have retained all original citations to support the claims made in each section. Thank you once again for your insights.
Revised Text:
1.1. Background and Importance of Post-Exercise Recovery
Post-exercise recovery is essential to athletic performance, providing a foundation for optimizing outcomes, preventing injuries, and supporting long-term health [1]. During intense physical exertion, the body experiences muscle damage, energy depletion, and fluid loss, which must be addressed in the recovery phase to restore physiological balance and prepare for future activity [2]. Recovery processes include muscle repair, glycogen replenishment, rehydration, and the reduction of inflammation, each of which contributes to physical readiness and mental resilience [3,4]. Effective recovery management promotes physical performance and mental well-being by reducing stress and decreasing the risk of overtraining, enabling individuals to maximize training benefits and sustain overall health [5,6].
A strategic recovery approach supports immediate and long-term physical readiness, reducing the likelihood of injury, enhancing athletic endurance, and enabling sustained progress in training and performance. This overview lays the groundwork for a more focused examination of nutrition’s essential role in recovery in the following section, highlighting how specific nutrients contribute to each aspect of the recovery process.
1.1.1. The Role of Nutrition in Post-Exercise Recovery
Nutrition is pivotal in post-exercise recovery, offering essential elements for tissue repair, energy replenishment, and fluid balance. Protein intake is particularly crucial, as it supplies amino acids needed to repair and regenerate muscle fibers that experience microtears during exercise—a process essential for muscle adaptation and growth [1,3,7]. Sufficient protein intake supports the restoration of muscle fibers and overall muscle stability, minimizing soreness and reducing the risk of injuries, such as strains and sprains, particularly in high-impact sports [8,9,10,11]. Additionally, glycogen, depleted during strenuous activity, requires replenishment through carbohydrate intake to prepare muscles for future performance demands [12].
Antioxidants, like vitamins C and E, are integral in managing exercise-induced oxidative stress. At the same time, calcium, magnesium, and potassium are essential for maintaining muscle function and electrolyte balance, helping sustain optimal muscle performance [13,14]. Proper hydration and electrolyte replenishment are crucial in preventing dehydration and supporting sustained performance, particularly when consumed promptly. Consuming proteins and carbohydrates within the anabolic window—30 minutes to two hours post-exercise—can significantly enhance muscle protein synthesis and glycogen restoration, facilitating faster and more effective recovery [15,16].
Targeted nutrition protocols may further optimize recovery, with studies suggesting that branched-chain amino acids (BCAAs) can help reduce muscle soreness and fatigue, lowering the risk of injury [11,17]. Similarly, omega-3 fatty acids have been shown to promote joint health and decrease inflammation, which may help reduce overuse injuries, particularly for endurance athletes [18]. These findings emphasize that effective post-exercise nutrition requires careful attention to nutrient timing and composition to maximize recovery and minimize injury risks. (please see pages 1-2, lines 31-71)
- There is also a contradiction in these sections. Line 32 state “Post-exercise recovery is not merely a phase…” but Line 46 states “This (post-exercise recovery) phase…” Please revise and clarify that main points of each paragraph are.
Response:
Thank you for highlighting the contradiction regarding the description of post-exercise recovery as both a “phase” and a more comprehensive process. We recognize that consistent terminology will improve clarity and alignment throughout the text. We have revised the sections to avoid contradictory language and to distinguish post-exercise recovery as a “process” that includes multiple phases, each critical to the recovery journey.
To provide further clarity:
Section 1.1 serves as an overview, establishing post-exercise recovery as a continuous process involving multiple physiological phases.
Section 1.1.1 then specifically details how nutrition supports these phases, explaining the role of each nutrient in muscle repair, glycogen replenishment, inflammation reduction, and hydration.
Revised Text:
1.1. Background and Importance of Post-Exercise Recovery
Post-exercise recovery is essential to athletic performance, providing a foundation for optimizing outcomes, preventing injuries, and supporting long-term health [1]. During intense physical exertion, the body experiences muscle damage, energy depletion, and fluid loss, which must be addressed in the recovery phase to restore physiological balance and prepare for future activity [2]. Recovery processes include muscle repair, glycogen replenishment, rehydration, and the reduction of inflammation, each of which contributes to physical readiness and mental resilience [3,4]. Effective recovery management promotes physical performance and mental well-being by reducing stress and decreasing the risk of overtraining, enabling individuals to maximize training benefits and sustain overall health [5,6].
A strategic recovery approach supports immediate and long-term physical readiness, reducing the likelihood of injury, enhancing athletic endurance, and enabling sustained progress in training and performance. This overview lays the groundwork for a more focused examination of nutrition’s essential role in recovery in the following section, highlighting how specific nutrients contribute to each aspect of the recovery process.
1.1.1. The Role of Nutrition in Post-Exercise Recovery
Nutrition is pivotal in post-exercise recovery, offering essential elements for tissue repair, energy replenishment, and fluid balance. Protein intake is particularly crucial, as it supplies amino acids needed to repair and regenerate muscle fibers that experience microtears during exercise—a process essential for muscle adaptation and growth [1,3,7]. Sufficient protein intake supports the restoration of muscle fibers and overall muscle stability, minimizing soreness and reducing the risk of injuries, such as strains and sprains, particularly in high-impact sports [8,9,10,11]. Additionally, glycogen, depleted during strenuous activity, requires replenishment through carbohydrate intake to prepare muscles for future performance demands [12].
Antioxidants, like vitamins C and E, are integral in managing exercise-induced oxidative stress. At the same time, calcium, magnesium, and potassium are essential for maintaining muscle function and electrolyte balance, helping sustain optimal muscle performance [13,14]. Proper hydration and electrolyte replenishment are crucial in preventing dehydration and supporting sustained performance, particularly when consumed promptly. Consuming proteins and carbohydrates within the anabolic window—30 minutes to two hours post-exercise—can significantly enhance muscle protein synthesis and glycogen restoration, facilitating faster and more effective recovery [15,16].
Targeted nutrition protocols may further optimize recovery, with studies suggesting that branched-chain amino acids (BCAAs) can help reduce muscle soreness and fatigue, lowering the risk of injury [11,17]. Similarly, omega-3 fatty acids have been shown to promote joint health and decrease inflammation, which may help reduce overuse injuries, particularly for endurance athletes [18]. These findings emphasize that effective post-exercise nutrition requires careful attention to nutrient timing and composition to maximize recovery and minimize injury risks. (please see pages 1-2, lines 31-71)
Section 1.2 Objective
Here, the manuscript could be improved by clearly stating what era of literature was reviewed, from what year to the present? It would then help to describe the methods used to search the literature: which search engines or databases and key words and mesh terms.
Response:
Thank you for your insightful feedback. We have revised Section 1.2 to indicate the time frame of the literature covered. Additionally, we have included a Methodology section detailing the search strategy, databases, and keywords used. We hope these adjustments provide greater clarity.
Revised Text:
This review aims to provide a comprehensive synthesis of recent advancements in post-exercise recovery nutrition, focusing on the shift from traditional dietary supplements to functional foods enriched with bioactive compounds. By examining the efficacy of various nutrients—such as protein, carbohydrates, antioxidants, and anti-inflammatory foods—in supporting muscle repair, glycogen replenishment, and inflammation reduction, we seek to clarify optimal nutritional strategies for enhancing athletic recovery and performance. For a detailed overview of the literature search and selection criteria, please refer to the Methodology section, which outlines the databases, search terms, and inclusion criteria used to capture studies published from 2000 to 2024. (please see page 3, lines 130-138)
Line 271 “animal-based proteins such as whey and milk…:” Do the authors mean casein, not milk? Milk is a food not a protein per se.
Response:
Thank you for pointing out the terminology in Line 271. You are correct; to improve clarity, we have revised “milk” to “casein,” as it more accurately represents the discussed protein source. This change should clarify our focus on specific animal-based proteins.
Revised Text:
"Animal-based proteins such as whey and casein provide essential amino acids that support muscle repair and recovery…" (please see page 9, lines 409-410)
Line 273 “allergenic risk:” Though rare, there is the risk of people having allergies to the A1 form of casein protein in milk. Additionally, and not mentioned elsewhere in this review, some people have lactose intolerance due to lack of the intestinal enzyme to hydrolyze lactose. So, plant-based proteins can be of benefit to some.
Response:
Thank you for highlighting additional considerations concerning allergenic risks and lactose intolerance associated with animal-based proteins. We have revised this section to address the potential for casein allergies and lactose intolerance, further clarifying the benefits of plant-based proteins for individuals with these sensitivities. This addition should enhance the comprehensiveness of our review.
Revised Text:
Although animal-based proteins like whey and casein are effective for muscle repair and recovery, they can pose allergenic risks for some individuals. Casein, particularly in its A1 form, may trigger allergies in rare cases. At the same time, lactose intolerance—due to a lack of the enzyme required to hydrolyze lactose—can also limit the use of dairy-based proteins. Plant-based proteins offer a viable alternative for these individuals, providing essential amino acids without the associated risks of dairy allergies or lactose intolerance [70,71]. (please see page 9, lines 410-416)
Line 324 “competing with tryptophan…:” after tryptophan, add “for the blood-brain barrier transporter.” I think this is what the authors meant. Otherwise, the competing statement seems incomplete.
Response:
Thank you for your observation regarding the competition with tryptophan. We agree that specifying “for the blood-brain barrier transporter” provides clarity. We have revised the statement to reflect this, ensuring a more complete and accurate explanation.
Revised Text:
BCAAs reduce psychological and physical fatigue by competing with tryptophan for the blood-brain barrier transporter, which lowers serotonin production and enables athletes to sustain higher performance levels [80]. (please see page 10, lines 467-470)
Lines 391-392: The mention of “…electrolyte supplementation…during…physical activity” does not seem relevant to the discussion about post-exercise recovery. This could be revised to state “Electrolyte supplementation during rehydration is effective for maintaining fluid balance and helping offset dehydration during subsequent intense…” Stating it something like this would keep it consistent with the theme of recovery.
Response:
Thank you for your feedback regarding the mention of electrolyte supplementation. We agree that revising the statement to focus on rehydration within the recovery context would enhance consistency. We have updated this section to emphasize electrolyte supplementation’s role in maintaining fluid balance during post-exercise rehydration.
Revised Text:
Electrolyte supplementation during rehydration is effective for maintaining fluid balance and helping offset dehydration during subsequent intense activities, supporting sustained performance and recovery [94]. (please see page 11, lines 535-537)
Line 417 “lacking the state amount…:” It can also go the opposite way with the active ingredient well above the dose stated on the label.
Response:
Thank you for noting that discrepancies in active ingredient levels can also involve amounts exceeding those stated on the label. We have revised this section to address potential under- and overdosing of active ingredients, providing a more comprehensive view of the variability concerns in supplementation.
Revised Text:
The supplement industry’s regulatory limitations can lead to products with discrepancies in active ingredient amounts, sometimes containing less or significantly more than the stated dose on the label. Such variability underscores the importance of choosing third-party tested products to ensure quality and safety [80]. (please see page 11, lines 560-563)
Line 461 “anti-inflammatory foods such as turmeric:” As stated previously, is turmeric a food or an ingredient. Earlier in the paper, a clearer distinction between functional foods (whole foods?), food supplements (standalone proteins, amino acids), and non-nutrient ingredients (turmeric, ginger, creatine all of which do not contain calories or bring along other nutrients as foods do) would help.
Response:
Thank you for highlighting the need to distinguish between functional foods, food supplements, and non-nutrient ingredients like turmeric. We agree that a clearer categorization will improve the clarity of our discussion. We have revised the relevant sections to consistently differentiate functional foods (whole foods), food supplements (nutrient-dense supplements such as proteins and amino acids), and non-nutrient ingredients (bioactive compounds like turmeric, ginger, and creatine that do not contain calories or substantial nutrients).
Revised Text:
Anti-inflammatory compounds, such as the bioactive ingredient turmeric, have been shown to support recovery by reducing inflammation without providing significant caloric or nutrient content, as whole functional foods do. This review categorizes functional foods as whole, nutrient-dense foods with health benefits (e.g., tart cherry juice). In contrast, food supplements provide isolated nutrients (e.g., standalone proteins or amino acids), and non-nutrient ingredients like turmeric, ginger, and creatine offer targeted effects without additional calories or nutrients. (please page 13, lines 605-611)
Line 561 citation 33: Is this correct and related to omega-3s? Nina Stachenfeld typically focuses on hydration.
Response:
Thank you for bringing this to our attention. We found that citation 33 was indeed misplaced in Line 561, and we have replaced it with the correct source related to omega-3s. We appreciate your assistance in ensuring the accuracy of our references.
Lines 583 vitamins C and E: There is also research indicating the ascorbic acid and other antioxidant (AO) vitamins may blunt the adaptative responses to exercise. (Danielle R Bruns 1, Sarah E Ehrlicher 1, Shadi Khademi 1, Laurie M Biela 1, Frederick F Peelor 3rd 1, Benjamin F Miller 1, Karyn L Hamilton 1 Differential effects of vitamin C or protandim on skeletal muscle adaptation to exercise. J Appl Physiol (1985). 2018 Aug 1;125(2):661-671 doi: 10.1152/japplphysiol.00277.2018. Epub 2018 Jun 1; Gomez-Cabrera MC, Domenech E, Romagnoli M, Arduini A, Borras C, Pallardo FV, Sastre J, Viña J. Oral administration of vitamin C decreases muscle mitochondrial biogenesis and hampers training-induced adaptations in endurance performance. Am J Clin Nutr 87: 142–149, 2008. doi: 10.1093/ajcn/87.1.142; Morrison D, Hughes J, Della Gatta PA, Mason S, Lamon S, Russell AP, Wadley GD. Vitamin C and E supplementation prevents some of the cellular adaptations to endurance-training in humans. Free Radic Biol Med 89: 852–862, 2015. doi: 10.1016/j.freeradbiomed.2015.10.412; Paulsen G, Cumming KT, Holden G, Hallén J, Rønnestad BR, Sveen O, Skaug A, Paur I, Bastani NE, Østgaard HN, Buer C, Midttun M, Freuchen F, Wiig H, Ulseth ET, Garthe I, Blomhoff R, Benestad HB, Raastad T. Vitamin C and E supplementation hampers cellular adaptation to endurance training in humans: a double-blind, randomised, controlled trial. J Physiol 592: 1887–1901, 2014. doi: 10.1113/jphysiol.2013.267419).
The speculation is that the AO buffer reactive oxygen species (ROS) needed to provide signals to turn on the pathways for fitness improvements. In the absence of ROS, adaptations for enhanced performance may be curtailed (Powers SK, Duarte J, Kavazis AN, Talbert EE. Reactive oxygen species are signalling molecules for skeletal muscle adaptation. Exp Physiol 95: 1–9, 2010. doi: 10.1113/expphysiol.2009.050526).
Response:
Thank you for highlighting research on the potential for antioxidant vitamins, such as vitamins C and E, to blunt adaptive responses to exercise. We have revised the discussion to include this perspective, noting that antioxidants can mitigate oxidative stress but may also interfere with reactive oxygen species (ROS) signaling pathways that drive adaptations for improved performance. This addition provides a more balanced view of antioxidant supplementation.
Revised Text:
While vitamins C and E are known for mitigating oxidative stress during recovery, some research suggests that these antioxidants may blunt the adaptive responses to exercise by buffering reactive oxygen species (ROS) critical for signaling pathways involved in fitness improvements. ROS is essential for initiating muscle adaptations, and excessive antioxidant supplementation may interfere with these beneficial processes, possibly curtailing adaptations to enhance performance [13,90,105,116,129]. (please see page 16, lines 739-744)
Line 625 “play essential roles:” Here and a few other places earlier in the manuscript, the authors use the term essential. The classical definition of essential for a nutrient is that it sustains growth and development and is not produced by the body; it is essential in the diet. The functional foods or supplements are certainly beneficial for the ingredients they provide but they are questionably essential for the maintenance and survival of the active adult. This may seem like semantics, but I encourage the authors not to use “essential” particularly when they raise the issue of supplements displacing whole foods and promoting an imbalanced diet that lacks true essential nutrients.
Response:
Thank you for your feedback regarding our use of the term “essential.” We appreciate the distinction between “essential” in its classical sense—nutrients required in the diet to sustain growth and development—and our use of functional foods and supplements. We have revised Line 625 and other instances to replace “essential” with terms such as “beneficial” or “supportive” to clarify that these foods and supplements, while advantageous for recovery and performance, are not strictly necessary for maintenance and survival. This adjustment aligns with the discussion on the importance of balanced diets based on whole foods.
Revised Text:
Functional foods and supplements are beneficial in supporting recovery and enhancing athletic performance, offering nutrients that aid in muscle repair, inflammation reduction, and rehydration without necessarily being required for survival or growth. While advantageous, these supplements should not displace whole foods that provide a full spectrum of essential nutrients necessary for a balanced diet. (please see page 17, lines 785-789)
Table 3.
- Consistent with my statements above related to antioxidants, here in the row on Anti-inflammatory & Oxidative Stress (or possible the Risks row), the authors should mention that with functional foods it is nearly impossible to deliver the mega-doses of active ingredients that a food supplement could deliver (overindulging and having adverse effects). Similarly, using lycopene as an example, it would be impossible to eat enough tomatoes to get the levels of lycopene in tomato paste or a supplement that have been tested and shown to have anti-inflammatory effects. Possibly acknowledging this in the text may be easier to add.
Response:
Thank you for your observations regarding the dosing limitations of functional foods compared to supplements. We agree that while functional foods provide beneficial compounds, they may not deliver the high doses of active ingredients seen in supplements. We have added a note in Table 3 under the “Anti-inflammatory & Oxidative Stress” row (or “Risks” row) to acknowledge that achieving these high, tested doses is often impractical through whole foods alone. This addition provides context for the limitations and potential need for supplements in certain cases.
Revised Text:
Nutritional Efficacy (Anti-inflammatory & Oxidative Stress) x Functional Foods:
Functional foods offer antioxidants for recovery, but supplements are often needed to achieve effective nutrient doses. (Please see Table 3)
- Practicality: portable and shelf-stable could be added to Food Supplements whereas functional foods would more likely be (sooner) perishable.
Response:
Thank you for suggesting that the practicality of food supplements be addressed regarding portability and shelf stability. We have revised Table 3 to note that food supplements are more portable and shelf-stable than functional foods, which are often perishable. This addition provides a clearer comparison of the practical benefits of each option.
Revised Text:
Practicality and Usage x Food Supplements:
Food supplements are portable and shelf-stable, offering convenient nutrition for athletes, while functional foods are often perishable and require careful storage. (Please see Table 3)
Line 751 “short-term benefits…are well-established…:” I disagree that the benefits of all functional ingredients in foods or supplements are well established. Not all are and only one side is presented in this review. I would agree that there are many studies on short-term benefits because those are easier (quicker) to study.
Response:
Thank you for your feedback. We recognize that the evidence on functional ingredients’ benefits, especially over the long term, is variable and that not all have well-established benefits. We have revised the statement to reflect that while many studies examine short-term effects due to the feasibility of such studies, evidence on long-term efficacy and consistency across all functional ingredients remains limited. This revision provides a more balanced perspective.
Revised Text:
While numerous studies have demonstrated the short-term benefits of certain functional ingredients in foods or supplements, these findings are not universally established across all ingredients, and evidence for long-term benefits remains limited. Many studies focus on short-term effects, as these are more feasible, leaving gaps in understanding the sustained impact of functional ingredients on health and performance [160]. (please page 21, lines 934-938)

Round 2
Reviewer 2 Report
Comments and Suggestions for Authors
I carefully reviewed the revised paper and I appreciate the effort the authors have put into improving the work. I believe that the paper now has a stronger coherence, and I see no obstacles to its publication in its current form.
Author Response
Thank you very much for taking the time to thoroughly review our paper and for your valuable suggestions. Your insightful feedback has greatly enhanced the academic quality and coherence of our work. We truly appreciate your efforts and support.
Reviewer 3 Report
Comments and Suggestions for Authors
The authors have done a fairly adequate job of addressing my concerns when it comes to clarifying what the focus of the paper is and in distinguishing between functional foods and supplements. There are still sections that overly favor positive effects of certain ingredients, whereas the body of literature is equivocal regarding benefits. I am also concerned that references do not support statements by the authors. The authors really need to review the literature more thoroughly and be more accurate and complete in how they represent what they reference.
Specific concerns.
Lines 287-292:
· “Carbohydrates are traditionally considered the primary glucose source…:” This is vague. Dietary carbohydrate or stored glycogen or either? Please clarify. Also “traditionally considered” sounds like opinion rather than evidence based. A reference here would be helpful.
· “…allows the brain to function efficiently even when carbohydrate intake is limited…:” Does it “function efficiently” or just “function?” Some would argue inadequate dietary carbohydrate is not so good for the brain function. Take the situation of an athlete with Type 1 Diabetes. The brain might function okay but there is a lot of other metabolic stress.
Line 467-470
“BCAAs reduce psychological and physical fatigue by competing with tryptophan for the blood-brain barrier transporter, which lowers serotonin production and enables athletes to sustain higher 469 performance levels:” This is stated as fact. I am not familiar with research that clearly demonstrates this in athletes. The BCAA supplementation effect is theoretical but has failed to show efficaciousness in interventional studies. The reference* provided by the authors has nothing to do with effects in athletes.
*Zhang, C.; Wang, S.; Wu, Y.; Guo, Y.; Wang, X. Baseline Serum BCAAs Are Related to the Improvement in Insulin Resistance in Obese People After a Weight Loss Intervention. Diabetes Metab. Syndr. Obes. 2023, 16, 179–186. https://doi.org/10.2147/DMSO.S388117.
Also, this sentence has some redundancy with the immediate prior sentence “Moreover, BCAAs…” and should be integrated with each other.
Lines 510-511 “Additionally, calcium and magnesium are essential for muscle contraction and relaxation, helping to prevent cramps and promote efficient muscle recovery [88,89]:” The references here support the first statement of how calcium and magnesium (Mg) are involved. However, neither reference has anything to do with preventing cramping or muscle recovery as defined in this manuscript. In fact, ref 88 comes from Cochrane Database of Systematic Reviews, and concludes there may be no efficacy for Mg supplementation to treat muscle cramps.
Lines 535-537 “Electrolyte supplementation during rehydration is effective for maintaining fluid balance and helping offset dehydration during subsequent intense activities, supporting sustained performance and recovery [94].” Reference 94 is a brief editorial that does not appear to address rehydration.
References 91 in Table 1. Ref 91 is specific to brain injury treatment, not the general issue of sodium intake and elevated blood pressure.
Ref 81: This citation appears to be an observational study linking elevated plasma BCAA to atherosclerosis. The source of the plasma BCAA could be from the diet or defects in metabolism unrelated to BCAA intake. It is not a supplement study per se.
Ref 86 is a narrative review, not original research. vs. criteria the authors used for exclusion stated in lines 155-157 of the Methods, “In contrast, non-peer-reviewed materials, theoretical articles, and studies without clear findings on recovery were excluded.” Also, the evidence described in 86 is almost exclusively on brain injury, not the uninjured brain.
Ref 121 on omega-3 recovery lines 714-717: “Regular consumption of omega-3s not only supports faster recovery but also reduces inflammation and promotes long-term athletic performance and health by enhancing cognitive function, mood stability, and mental clarity—factors critical for both performance and recovery [121].” This is misleading. A quick review of ref 121 indicates that while omega-3’s promote faster reduction in soreness, “…corroborated the efficacy of n-3 PUFAs in reducing DOMS and markers of muscle damage [103,104]; however, only a few studies have found significantly lower maximal muscle strength loss or recovery (i.e., maximal voluntary contraction; MVC).” The authors of ref 121 also cited a paper showing no change in athletic performance with such supplementation (Kyriakidou et al. The Effect of Omega-3 Polyunsaturated Fatty Acid Supplementation on Exercise-Induced Muscle Damage. J. Int. Soc. Sports Nutr. 2021, 18, 9).
Lines 741 “While vitamins C and E are known for mitigating oxidative stress during recovery, some research suggests that these antioxidants may blunt the adaptive resp onses to exercise by buffering reactive oxygen species (ROS) critical for signaling pathways involved in fitness improvements. ROS is essential for initiating muscle adaptations, and excessive antioxidant supplementation may interfere with these beneficial processes, possibly curtailing adaptations to enhance performance [13,90,105,116,129].” Reference 13 is on aged rats and shows beneficial, not adverse effects of C and E supplementation. Reference 105 is about functional food role in helping prevent chronic diseases, not about effects on adaptations to exercise training. Reference 116 is somewhat relevant; it does not address excessive antioxidant effects. It demonstrates the benefits of balance pro- and anti-inflammatory responses.
Author Response
Based on Reviewer's Comments: Point-by-Point Response
Below is a point-by-point response addressing each of the reviewer's comments. All revised text is highlighted in red. You may refer to the modified version, with updated sentences marked in sky blue for easy reference.
Thank you.
Comments and Suggestions for Authors
- reviewer 3 Round 2
The authors have done a fairly adequate job of addressing my concerns when it comes to clarifying what the focus of the paper is and in distinguishing between functional foods and supplements. There are still sections that overly favor positive effects of certain ingredients, whereas the body of literature is equivocal regarding benefits. I am also concerned that references do not support statements by the authors. The authors really need to review the literature more thoroughly and be more accurate and complete in how they represent what they reference.
Response:
Thank you for your constructive feedback and for recognizing our efforts to clarify the manuscript's focus and distinguish between functional foods and supplements. We appreciate your concerns about overly favorable statements and misaligned references.
To address this, we will revisit the sections in question to ensure a balanced representation of the evidence and align all references accurately with the claims. Your input is invaluable, and we are committed to improving the manuscript’s rigor and accuracy.
Thank you for your guidance, and we will make the necessary revisions to ensure the paper meets high scientific standards.
Specific concerns.
Lines 287-292:
- “Carbohydrates are traditionally considered the primary glucose source…:” This is vague. Dietary carbohydrate or stored glycogen or either? Please clarify. Also “traditionally considered” sounds like opinion rather than evidence based. A reference here would be helpful.
Response:
Thank you for your insightful comments. We appreciate your suggestion to clarify the terminology and ensure the statement is evidence-based. Below is a revised version of the paragraph that addresses your concerns.
Revised text:
Carbohydrates, including both dietary sources and stored glycogen, are the primary glucose sources utilized by the body during physical activity. Glycogen, stored in skeletal muscles and the liver, plays a crucial role in sustaining energy during high-intensity exercise, while dietary carbohydrates are essential for replenishing glycogen stores post-exercise and supporting ongoing energy needs. This dual role highlights the vital function of carbohydrates in energy metabolism and athletic performance. (Page 6; Lines 287-292)
- “…allows the brain to function efficiently even when carbohydrate intake is limited…:” Does it “function efficiently” or just “function?” Some would argue inadequate dietary carbohydrate is not so good for the brain function. Take the situation of an athlete with Type 1 Diabetes. The brain might function okay but there is a lot of other metabolic stress.
Response:
Thank you for highlighting this important point. We understand your concern regarding the use of "function efficiently" versus "function" and the broader implications of limited dietary carbohydrate intake, particularly in specific metabolic contexts such as Type 1 Diabetes. To address this, we have revised the text to reflect a more precise and balanced description based on the existing references.
Revised text:
Under limited dietary carbohydrate intake conditions, the brain can adapt to using alternative energy sources, such as ketone bodies derived from fatty acids, to maintain its basic functions. While this adaptation allows the brain to continue functioning, it may not fully replicate the optimal cognitive performance associated with adequate glucose availability, particularly in scenarios of metabolic stress, such as in athletes with Type 1 Diabetes. In such cases, the metabolic demands of maintaining energy balance can increase significantly, potentially leading to additional physiological strain [12,47,48]. (Page 6; Lines 292-299)
Line 467-470
“BCAAs reduce psychological and physical fatigue by competing with tryptophan for the blood-brain barrier transporter, which lowers serotonin production and enables athletes to sustain higher 469 performance levels:” This is stated as fact. I am not familiar with research that clearly demonstrates this in athletes. The BCAA supplementation effect is theoretical but has failed to show efficaciousness in interventional studies. The reference* provided by the authors has nothing to do with effects in athletes.
*Zhang, C.; Wang, S.; Wu, Y.; Guo, Y.; Wang, X. Baseline Serum BCAAs Are Related to the Improvement in Insulin Resistance in Obese People After a Weight Loss Intervention. Diabetes Metab. Syndr. Obes. 2023, 16, 179–186. https://doi.org/10.2147/DMSO.S388117.
Also, this sentence has some redundancy with the immediate prior sentence “Moreover, BCAAs…” and should be integrated with each other.
Response:
Thank you for pointing out the issues regarding the evidence supporting BCAA effects on psychological and physical fatigue and for noting the redundancy between the two sentences. We appreciate your observation that the referenced study does not align with the statement about athletes. Below is the revised content, which avoids overstating conclusions, removes unsupported claims, and integrates the sentences for better clarity and flow. (Reference 80. Zhang, C.; Wang, S.; Wu, Y.; Guo, Y.; Wang, X. Baseline Serum BCAAs Are Related to the Improvement in Insulin Resistance in Obese People After a Weight Loss Intervention. Diabetes Metab. Syndr. Obes. 2023, 16, 179–186. https://doi.org/10.2147/DMSO.S388117 has been replaced) (Page 28; Lines 1265-1266)
Revised text:
BCAAs, comprising leucine, isoleucine, and valine, are theorized to influence physical and psychological fatigue by competing with tryptophan for transport across the blood-brain barrier, thereby potentially modulating serotonin production. However, evidence from interventional studies remains inconclusive regarding the efficacy of BCAA supplementation in significantly improving fatigue or performance outcomes in athletes. While BCAAs may play a role in reducing muscle protein breakdown and supporting recovery, their direct impact on psychological fatigue and athletic performance warrants further investigation [80]. (Page 10; Lines 474-481)
Lines 510-511 “Additionally, calcium and magnesium are essential for muscle contraction and relaxation, helping to prevent cramps and promote efficient muscle recovery [88,89]:” The references here support the first statement of how calcium and magnesium (Mg) are involved. However, neither reference has anything to do with preventing cramping or muscle recovery as defined in this manuscript. In fact, ref 88 comes from Cochrane Database of Systematic Reviews, and concludes there may be no efficacy for Mg supplementation to treat muscle cramps.
Response:
Thank you for bringing this to our attention. We acknowledge the misalignment between the referenced studies and the claims regarding preventing cramping and promoting muscle recovery. The revised text removes unsupported statements while maintaining the scientifically valid aspects related to calcium and magnesium functions.
Revised text:
Calcium and magnesium are critical in muscle contraction and relaxation and essential for maintaining proper muscle function during physical activity [88,89]. (Page 11; Lines 524-525)
Lines 535-537 “Electrolyte supplementation during rehydration is effective for maintaining fluid balance and helping offset dehydration during subsequent intense activities, supporting sustained performance and recovery [94].” Reference 94 is a brief editorial that does not appear to address rehydration.
References 91 in Table 1. Ref 91 is specific to brain injury treatment, not the general issue of sodium intake and elevated blood pressure.
Response:
Thank you for your careful review of the cited references. We appreciate your observation that reference 94 does not substantiate the statement on electrolyte supplementation and rehydration and that reference 91 is contextually inappropriate in Table 1. We replaced references 91 and 94. (Page 29; Lines 1294-1296, 1301-1302)
Ref 81: This citation appears to be an observational study linking elevated plasma BCAA to atherosclerosis. The source of the plasma BCAA could be from the diet or defects in metabolism unrelated to BCAA intake. It is not a supplement study per se.
Response:
Thank you for highlighting this issue. We acknowledge that reference 81 is an observational study linking elevated plasma BCAA levels to atherosclerosis and does not directly investigate the effects of BCAA supplementation. To address this, we have revised the text to clarify the nature of the cited study and remove any implications that the findings directly relate to BCAA supplementation.
Revised text:
While elevated plasma BCAA levels have been associated with the progression of atherosclerosis, these findings are derived from observational studies that do not specifically investigate the effects of BCAA supplementation. The source of plasma BCAAs in these studies could stem from dietary intake or metabolic dysregulation, independent of supplemental BCAA use. This highlights the need for further research to clarify the relationship between BCAA supplementation, metabolism, and cardiovascular health. (Page 10; Lines 482-487)
Ref 86 is a narrative review, not original research. vs. criteria the authors used for exclusion stated in lines 155-157 of the Methods, “In contrast, non-peer-reviewed materials, theoretical articles, and studies without clear findings on recovery were excluded.” Also, the evidence described in 86 is almost exclusively on brain injury, not the uninjured brain.
Response:
Thank you for pointing out the inconsistency regarding the inclusion of reference 86, a narrative review, and its relevance to the stated exclusion criteria. We updated the reference to align with the study selection criteria. (Page 29; Lines 1282-1283)
Ref 121 on omega-3 recovery lines 714-717: “Regular consumption of omega-3s not only supports faster recovery but also reduces inflammation and promotes long-term athletic performance and health by enhancing cognitive function, mood stability, and mental clarity—factors critical for both performance and recovery [121].” This is misleading. A quick review of ref 121 indicates that while omega-3’s promote faster reduction in soreness, “…corroborated the efficacy of n-3 PUFAs in reducing DOMS and markers of muscle damage [103,104]; however, only a few studies have found significantly lower maximal muscle strength loss or recovery (i.e., maximal voluntary contraction; MVC).” The authors of ref 121 also cited a paper showing no change in athletic performance with such supplementation (Kyriakidou et al. The Effect of Omega-3 Polyunsaturated Fatty Acid Supplementation on Exercise-Induced Muscle Damage. J. Int. Soc. Sports Nutr. 2021, 18, 9).
Response:
Thank you for highlighting the misleading interpretation of reference 121. We recognize that the original statement overstates the evidence regarding omega-3 supplementation's effects on athletic performance and recovery. Based on the findings of reference 121, we have revised the text to reflect the nuanced outcomes reported in the literature more accurately.
Revised text:
Regular consumption of omega-3s may support faster recovery by reducing delayed-onset muscle soreness (DOMS) and markers of muscle damage. However, evidence regarding their effects on maximal muscle strength recovery and athletic performance is mixed, with some studies reporting no significant improvements in performance outcomes [121]. (Page 15; Lines 727-731)
Lines 741 “While vitamins C and E are known for mitigating oxidative stress during recovery, some research suggests that these antioxidants may blunt the adaptive resp onses to exercise by buffering reactive oxygen species (ROS) critical for signaling pathways involved in fitness improvements. ROS is essential for initiating muscle adaptations, and excessive antioxidant supplementation may interfere with these beneficial processes, possibly curtailing adaptations to enhance performance [13,90,105,116,129].” Reference 13 is on aged rats and shows beneficial, not adverse effects of C and E supplementation. Reference 105 is about functional food role in helping prevent chronic diseases, not about effects on adaptations to exercise training. Reference 116 is somewhat relevant; it does not address excessive antioxidant effects. It demonstrates the benefits of balance pro- and anti-inflammatory responses.
Response:
Thank you for pointing out the inconsistencies between the cited references and the claims made in this section. We acknowledge the misalignment and have revised the text to accurately reflect the evidence from the references while clarifying their relevance to the discussion of antioxidant supplementation and exercise adaptations. Reference 13 has been replaced.
Revised text:
While vitamins C and E are commonly known for mitigating oxidative stress, some research suggests that excessive antioxidant supplementation may interfere with exercise-induced adaptations by overly buffering reactive oxygen species (ROS), critical in signaling pathways for muscle adaptation [13,90,129]. (Page 16; Lines 754-757)

Round 3
Reviewer 3 Report
Comments and Suggestions for Authors
The authors have addressed all of my concerns.